# RANDOM LAPLACIAN FEATURES FOR LEARNING WITH HYPERBOLIC SPACE

## ABSTRACT

Due to its geometric properties, hyperbolic space can support high-fidelity embeddings of tree- and graph-structured data, upon which various hyperbolic networks have been developed. Existing hyperbolic networks encode geometric priors not only for the input, but also at every layer of the network. This approach involves repeatedly mapping to and from hyperbolic space, which makes these networks complicated to implement, computationally expensive to scale, and numerically unstable to train. In this paper, we propose a simpler approach: learn a hyperbolic embedding of the input, then map once from it to Euclidean space using a mapping that encodes geometric priors by respecting the isometries of hyperbolic space, and finish with a standard Euclidean network. The key insight is to use a random feature mapping via the eigenfunctions of the Laplace operator, which we show can approximate any isometry-invariant kernel on hyperbolic space. Our method can be used together with any graph neural networks: using even a linear graph model yields significant improvements in both efficiency and performance over other hyperbolic baselines in both transductive and inductive tasks.

## 1 INTRODUCTION

Real-world data contains various structures that resemble non-Euclidean spaces: for example, data with tree- or graph-structure such as citation networks (Sen et al., 2008), social networks (Hoff et al., 2002), biological networks (Rossi & Ahmed, 2015), and natural language (e.g., taxonomies and lexical entailment) where latent hierarchies exist (Nickel & Kiela, 2017). Graph-style data features in a range of problems—including node classification, link prediction, relation extraction, and text classification. It has been shown both theoretically and empirically (Bowditch, 2006; Nickel & Kiela, 2017; 2018; Chien et al., 2022) that hyperbolic space—the geometry with constant negative curvature—is naturally suited for representing (i.e. embedding) such data and capturing implicit hierarchies, outperforming Euclidean baselines. For example, Sala et al. (2018) shows that hyperbolic space can embed trees without loss of information (arbitrarily low distortion), which cannot be achieved by Euclidean space of any dimension (Chen et al., 2013; Ravasz & Barabási, 2003).

Presently, most well-known and -established deep neural networks are built in Euclidean space. The standard approach is to pass the input to a Euclidean network and hope the model can learn the features and embeddings. But this flat-space approach can encode the wrong prior in tasks for which we know the underlying data has a different geometric structure, such as the hyperbolic-space structure implicit in tree-like graphs. Motivated by this, there is an active line of research on developing ML models in hyperbolic space $\mathbb{H}_n$. Starting from hyperbolic neural networks (HNN) by Ganea et al. (2018), a variety of hyperbolic networks were proposed for different applications, including HNN++ (Shimizu et al., 2020), hyperbolic variational auto-encoders (HVAE, Mathieu et al. (2019)), hyperbolic attention networks (HATN, Gulcehre et al. (2018)), hyperbolic graph convolutional networks (HGCN, Chami et al. (2019)), hyperbolic graph neural networks (HGNN, Liu et al. (2019)), and hyperbolic graph attention networks (HGAT Zhang et al. (2021a)). The strong empirical results of HGCN and HGNN in particular on node classification, link prediction, and molecular-and-chemical-property prediction show the power of hyperbolic geometry for graph learning.

These hyperbolic networks adopt hyperbolic geometry at every layer of the model. Since hyperbolic space is not a vector space, operations such as addition and multiplication are not well-defined; neither are matrix-vector multiplication and convolution, which are key components of a deep model

that uses hyperbolic geometry at every layer. A common solution is to treat hyperbolic space as a *gyro-vector space* by equipping it with a non-commutative, non-associative addition and multiplication, allowing hyperbolic points to be processed as features in a neural network forward. However, this complicates the use of hyperbolic geometry in neural networks because the imposition of an extra structure on hyperbolic space beyond its manifold properties—making the approach somehow non-geometric. A second problem with using hyperbolic points as intermediate features is that these points can stray far from the origin (just as Euclidean DNNs require high dynamic range Kalamkar et al. (2019)), especially for deeper networks. This can cause significant numerical issues when the space is represented with ordinary floating-point numbers: the representation error is unbounded and grows exponentially with the distance from the origin. Much careful hyperparameter tuning is required to avoid this "NaN problem" Sala et al. (2018); Yu & De Sa (2019; 2021). These issues call for a simpler and more principled way of using hyperbolic geometry in DNNs.

In this paper, we propose such a simple approach for learning with hyperbolic space. The insight is to (1) encode the hyperbolic geometric priors *only at the input* via an embedding into hyperbolic space, which is then (2) mapped once into Euclidean space by a random feature mapping $\phi : \mathbb{H}_n \rightarrow \mathbb{R}^d$ that (3) respects the geometry of hyperbolic space in that its induced kernel $k(\boldsymbol{x}, \boldsymbol{y}) = \mathbb{E}[\langle \phi(\boldsymbol{x}), \phi(\boldsymbol{y}) \rangle]$ is *isometry-invariant*, i.e. $k(\boldsymbol{x}, \boldsymbol{y})$ depends only on the hyperbolic distance between $\boldsymbol{x}$ and $\boldsymbol{y}$, followed by (4) passing these Euclidean features through some downstream Euclidean network. This approach both avoids the numerical issues common in previous approaches (since hyperbolic space is only used once early in the network, numerical errors will not compound) and eschews the need for augmenting hyperbolic space with any additional non-geometric structure (since we base the mapping only on geometric distances in hyperbolic space). Our contributions are as follows:

- In Section 4 we propose a random feature extraction called HyLa which can be sampled to be an unbiased estimator of any isometry-invariant kernel on hyperbolic space. This generalizes the classic method of random Fourier features proposed for Euclidean space by Rahimi et al. (2007).

- In Section 5 we show how to adopt HyLa in an end-to-end graph learning architecture that simultaneously learns the embedding of the initial objects and the Euclidean graph learning model.

- In Section 6, we evaluate our approach empirically. Our HyLa-networks demonstrate better performance, scalability and computation speed than existing hyperbolic networks: HyLa-networks consistently outperform HGCN, even on a tree dataset, with 12.3% improvement while being 4.4× faster. Meanwhile, we argue that our method is an important hyperbolic baseline to compare against due to its simple implementation and compatibility with any graph learning model.

## 2 RELATED WORK

**Hyperbolic space.** $n$-dimensional hyperbolic space $\mathbb{H}_n$ is usually defined and used via a model, a representation of $\mathbb{H}_n$ within Euclidean space. Common choices include the Poincaré ball (Nickel & Kiela, 2017) and Lorentz hyperboloid model (Nickel & Kiela, 2018). We develop our approach using the Poincaré ball model, but our methodology is independent of the model and can be applied to other models. The Poincaré ball model is the Riemannian manifold $(\mathcal{B}^n, g_p)$ with $\mathcal{B}^n = \{\boldsymbol{x} \in \mathbb{R}^n : \|\boldsymbol{x}\| < 1\}$ being the open unit ball and the Riemannian metric $g_p$ and metric distance $d_p$ being

$$g_p(\boldsymbol{x}) = 4(1 - \|\boldsymbol{x}\|^2)^{-2} g_e \qquad \text{and} \qquad d_p(\boldsymbol{x}, \boldsymbol{y}) = \operatorname{arcosh}\left(1 + 2\frac{\|\boldsymbol{x} - \boldsymbol{y}\|^2}{(1 - \|\boldsymbol{x}\|^2)(1 - \|\boldsymbol{y}\|^2)}\right).$$

where $g_e$ is the Euclidean metric. To encode geometric priors into neural networks, many versions of *hyperbolic neural networks* have been proposed. But while (matrix-) addition and multiplication are essential to develop a DNN, hyperbolic space is not a vector space with well-defined addition and multiplication. To handle this issue, several approaches were proposed in the literature.

**Gyrovector space.** Many hyperbolic networks, including HNN (Ganea et al., 2018), HNN++ (Shimizu et al., 2020), HVAE (Mathieu et al., 2019), HGAT (Zhang et al., 2021a), and GIL Zhu et al. (2020), adopt the framework of *gyrovector space* as an algebraic formalism for hyperbolic geometry, by equipping hyperbolic space with non-associative addition and multiplication: Möbius addition $\oplus$ and Möbius scalar multiplication $\otimes$, which is defined for $\boldsymbol{x}, \boldsymbol{y} \in \mathcal{B}^n$ and a scalar $r \in \mathbb{R}$

$$\boldsymbol{x} \oplus \boldsymbol{y} := \frac{(1 + 2\langle \boldsymbol{x}, \boldsymbol{y} \rangle + \|\boldsymbol{y}\|^2)\boldsymbol{x} + (1 - \|\boldsymbol{x}\|^2)\boldsymbol{y}}{1 + 2\langle \boldsymbol{x}, \boldsymbol{y} \rangle + \|\boldsymbol{x}\|^2 \|\boldsymbol{y}\|^2}, \quad r \otimes \boldsymbol{x} := \tanh(r \tanh^{-1}(\|\boldsymbol{x}\|)) \frac{\boldsymbol{x}}{\|\boldsymbol{x}\|}.$$

However, Möbius addition and multiplication are complicated with a high computation cost; high level operations such as Möbius matrix-vector multiplication are even more complicated and numer-

ically unstable (Yu & De Sa, 2021; Yu et al., 2022), due to the use of ill-conditioned functions like $\tanh^{-1}$. Also problematic is the way hyperbolic space is treated as a gyrovector space rather than a manifold, meaning this approach can not be generalized to other manifolds that lack this structure.

**Push-forward & Pull-backward**. Since many operations are well-defined in Euclidean space but not in hyperbolic space, a natural idea is to map $\mathbb{H}_n$ to $\mathbb{R}^d$ via some mappings, apply well-defined operations in $\mathbb{R}^d$, then map the results back to hyperbolic space. Many works, including HATN (Gulcehre et al., 2018), HGCN (Chami et al., 2019), HGNN (Liu et al., 2019), HGAT (Zhang et al., 2021a), Lorentzian GCN (Zhang et al., 2021b), and GIL Zhu et al. (2020), adopt this method:

- pull the hyperbolic points to Euclidean space with a "pull-backward" mapping;
- apply operations such as multiplication and convolution in Euclidean space; and then
- push the resulting Euclidean points to hyperbolic space with a "push-forward" mapping.

Since hyperbolic space and Euclidean space are different spaces, no isomorphic maps exist between them. A common choice of the mappings (Chami et al., 2019; Liu et al., 2019) is the exponential map $\exp_{\boldsymbol{x}}(\cdot)$ and logarithm map $\log_{\boldsymbol{x}}(\cdot)$, where $\boldsymbol{x}$ is usually chosen to be the origin $\boldsymbol{O}$. The exponential and logarithm maps are mappings between the hyperbolic space and its tangent space $\mathcal{T}_{\boldsymbol{x}}\mathbb{H}_n$, which is an Euclidean space contains gradients. The exponential map maps a vector $\boldsymbol{v} \in \mathcal{T}_{\boldsymbol{x}}\mathbb{H}_n$ at $\boldsymbol{x}$ to another point $\exp_{\boldsymbol{x}}(\boldsymbol{v})$ in $\mathbb{H}_n$ (intuitively, $\exp_{\boldsymbol{x}}(\boldsymbol{v})$ is the point reached by starting at $\boldsymbol{x}$ and moving in the direction of $\boldsymbol{v}$ a distance $\|\boldsymbol{v}\|$ along the manifold), while the logarithm map inverts this.

This approach is more straightforward and natural in the sense that hyperbolic space is only treated as a manifold object with no more structures added, so it can be generalized to general manifolds (although it does privilege the origin). However, $\exp_{\boldsymbol{o}}(\cdot)$ and $\log_{\boldsymbol{o}}(\cdot)$ are still complicated and numerically unstable. Both push-forward and pull-backward mappings are used at every hyperbolic layer, which incurs a high computational cost in both the model forward and backward loop. As a result, this prevents hyperbolic networks from scaling to large graphs. Moreover, the Push-forward & Pull-backward mappings act more like nonlinearities instead of producing meaningful features.

**Kernel Methods and Horocycle Features.** Cho et al. (2019) proposed hyperbolic kernel SVM for nonlinear classification without resorting to ill-fitting tools developed for Euclidean space. Their approach differs from ours in that they map hyperbolic points to another (higher-dimensional) hyperbolic feature space, rather than an Euclidean feature space. They also constructed feature mappings only for the Minkowski inner product kernel: it's unknown how to construct feature mappings of their type for general kernels. Another work by Fang et al. (2021) develops several valid positive definite kernels in hyperbolic spaces and investigates their usages; they do not provide any sampling-based features to approximate these kernels. Wang (2020) constructed hyperbolic neuron models using a push-forward mapping along with the *hyperbolic Poisson kernel* $P_n(\boldsymbol{x}, \boldsymbol{\omega}) = (\frac{1-\|\boldsymbol{x}\|^2}{\|\boldsymbol{x}-\boldsymbol{\omega}\|^2})^{n-1}$ for $\boldsymbol{x} \in \mathcal{B}^n$, $\boldsymbol{\omega} \in \partial\mathcal{B}^n$ as the backbone of an even more complicated feature function. Sonoda et al. (2022) theoretically proposes a continuous version of shallow fully-connected networks on non-compact symmetric space (including hyperbolic space) using Helgason-Fourier transform, where some network functions coincidentally share some similarities to features proposed in this paper.

## 3 BACKGROUND

**Laplace operator (Euclidean).** The Laplace operator $\Delta$ on Euclidean space $\mathbb{R}^n$ is defined as the divergence of the gradient of a scalar function $f$, i.e., $\Delta f = \nabla \cdot \nabla f = \sum_{i=1}^{n} \frac{\partial^2 f}{\partial x_i^2}$. The eigenfunctions of $\Delta$ are the solutions of the *Helmholtz equation* $-\Delta f = \lambda f, \lambda \in \mathbb{R}$, and can form an *orthonormal basis* for the Hilbert space $L^2(\Omega)$ when $\Omega \in \mathbb{R}^n$ is compact (Gilbarg & Trudinger, 2015), i.e., a linear combination of them can represent any function/model that is $L^2$-integrable. A notable parameterization for these eigenfunctions are the *plane waves*, given by $f(\boldsymbol{x}) = \exp(i\langle \boldsymbol{\omega}, \boldsymbol{x} \rangle)$, where $\boldsymbol{\omega} \in \mathbb{R}^n$ and $\langle \cdot, \cdot \rangle$ is the Euclidean inner product. A standard result given in Helgason (2022) states that any eigenfunction of $\Delta$ can be written as a linear combination of these plane waves (Theorem A.1). The famous result of Rahimi et al. (2007) used these eigenfunctions to construct feature maps, called *random Fourier features*, for arbitrary shift-invariant kernels in Euclidean space.

**Theorem 3.1** (Bochner's theorem, Rudin (2017))**.** *For any shift-invariant continuous kernel* $k(\boldsymbol{x}, \boldsymbol{y}) = k(\boldsymbol{x} - \boldsymbol{y})$ *on* $\mathbb{R}^n$, *let* $p(\boldsymbol{\omega})$ *be its Fourier transform and* $\xi_{\boldsymbol{\omega}}(\boldsymbol{x}) = \exp(i\langle \boldsymbol{\omega}, \boldsymbol{x} \rangle)$. *Then* $k$ *is positive definite if and only if* $p \geq 0$, *in which case if we draw* $\mathbf{w}$ *proportionally to* $p$,
$$k(\boldsymbol{x} - \boldsymbol{y}) = \int_{\mathbb{R}^n} p(\boldsymbol{\omega}) \exp(i\langle \boldsymbol{\omega}, \boldsymbol{x} - \boldsymbol{y} \rangle) \, d\boldsymbol{\omega} = k(\mathbf{0}) \cdot \mathbb{E}_{\boldsymbol{\omega} \sim p} [\xi_{\boldsymbol{\omega}}(\boldsymbol{x}) \xi_{\boldsymbol{\omega}}(\boldsymbol{y})^*].$$

Since both the probability distribution $p(\boldsymbol{\omega})$ and the kernel $k(\boldsymbol{x} - \boldsymbol{y})$ are real, the integral is unchanged when we replace the exponential with a cosine. Rahimi et al. (2007) leveraged this to produce real-valued features by setting $z_{\boldsymbol{\omega},b}(\boldsymbol{x}) = \sqrt{2}\cos(\langle\boldsymbol{\omega},\boldsymbol{x}\rangle + b)$, arriving at a real-valued mapping that satisfies the condition $\mathbb{E}_{\boldsymbol{w}\sim p, b\sim \mathrm{Unif}[0,2\pi]}\left[z_{\boldsymbol{\omega},b}(\boldsymbol{x})z_{\boldsymbol{\omega},b}(\boldsymbol{y})\right] = k(\boldsymbol{x} - \boldsymbol{y})$. Rahimi et al. (2007) approximated functions such as Gaussian, Laplacian and Cauchy kernels with this technique.

**Laplace-Beltrami operator (Hyperbolic).** The Laplace-Beltrami operator $\mathcal{L}$ is the generalization of the Laplace operator to Riemannian manifolds, defined as the divergence of the gradient for any twice-differentiable real-valued function $f$, i.e., $\mathcal{L}f = \nabla \cdot \nabla f$. In the $n$-dimensional Poincaré disk model $\mathcal{B}^n$, the Laplace-Beltrami operator takes the form (Agmon, 1987)

$\mathcal{L} = \frac{1}{4}(1 - \|\boldsymbol{x}\|^2)^2 \sum_{i=1}^n \frac{\partial^2}{\partial x_i^2} + \frac{n-2}{2}(1 - \|\boldsymbol{x}\|^2) \sum_{i=1}^n x_i \frac{\partial}{\partial x_i}$.

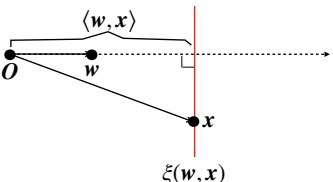

Just as in the Euclidean case, the eigenfunctions of $\mathcal{L}$ in hyperbolic space can be derived by solving the Helmholtz equation. We might hope to find analogs of the "plane waves" in hyperbolic space that are eigenfunctions of $\mathcal{L}$. One way to approach this is via a geometric interpretation of plane waves.

Figure 1: Euclidean hyperplane.

In the Euclidean case, for unit vector $\boldsymbol{\omega}$ and scalar $\lambda$, $f(\boldsymbol{x}) = \exp(i\lambda\langle\boldsymbol{\omega},\boldsymbol{x}\rangle)$ is called a "plane wave" because it is constant on each hyperplane perpendicular to $\boldsymbol{\omega}$. We can interpret $\langle\boldsymbol{\omega},\boldsymbol{x}\rangle$ as the signed distance from the origin $\boldsymbol{O}$ to the hyperplane $\xi(\boldsymbol{\omega},\boldsymbol{x})$ which contains $\boldsymbol{x}$ and is perpendicular to $\boldsymbol{w}$ (Figure 1).

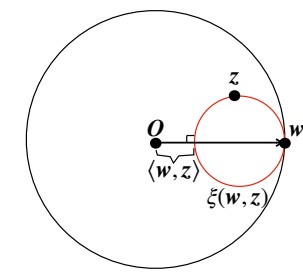

In the Poincaré ball model of hyperbolic space, the geometric analog of the hyperplane is the *horocycle*. For any $\boldsymbol{z} \in \mathcal{B}^n$ and unit vector $\boldsymbol{\omega}$ (i.e., $\boldsymbol{\omega} \in \partial\mathcal{B}^n$), the horocycle $\xi(\boldsymbol{\omega},\boldsymbol{z})$ is the Euclidean circle that passes through $\boldsymbol{\omega},\boldsymbol{z}$ and is tangent to the boundary $\partial\mathcal{B}^n$ at $\boldsymbol{\omega}$, as indicated in Figure 2. We let $\langle\boldsymbol{\omega},\boldsymbol{z}\rangle_H$ denote the signed hyperbolic distance from the origin

Figure 2: Hyperbolic horocycle.

$\boldsymbol{O}$ to the the horocycle $\xi(\boldsymbol{\omega},\boldsymbol{z})$. In the Poincaré ball model, this takes the form $\langle\boldsymbol{\omega},\boldsymbol{z}\rangle_H = \log((1-\|\boldsymbol{z}\|^2)/\|\boldsymbol{z}-\boldsymbol{\omega}\|^2)$. If we define the "hyperbolic plane waves" $\exp(\mu\langle\boldsymbol{\omega},\boldsymbol{z}\rangle_H)$, where $\mu \in \mathbb{C}$. Unsurprisingly, they are indeed eigenfunctions of the hyperbolic Laplacian (Agmon, 1987).

$$\mathcal{L}\exp(\mu\langle\boldsymbol{\omega},\boldsymbol{z}\rangle_H) = \mu(\mu - n + 1)e(\mu\langle\boldsymbol{\omega},\boldsymbol{z}\rangle_H).$$

Since we are interested in finding real eigenfunctions (via the same exp-to-cosine trick used in Rahimi et al. (2007)), we restrict our attention to $\mu$ that yield a real eigenvalue. This happens when $\mu = \frac{n-1}{2} + i\lambda$ for real $\lambda$, in which case the eigenvalue is $\mu(\mu - n + 1) = -\lambda^2 - (n-1)^2/4$. Just as the Euclidean plane waves $\exp(i\langle\boldsymbol{\omega},\boldsymbol{x}\rangle)$ span the eigenspaces of the Euclidean Laplacian, the same result holds for these "hyperbolic plane waves" (Theorem A.2).

## 4  HyLa: Euclidean Features from Hyperbolic Embeddings

In this section, we present HyLa, a feature mapping that can approximate an isometry-invariant kernel over hyperbolic space $\mathbb{H}_n$ in the same way that the random Fourier features of Rahimi et al. (2007) approximate any shift-invariant kernel over $\mathbb{R}^n$. In place of the Euclidean plain waves, which are the eigenfunctions of the Euclidean Laplacian, here we derive our feature extraction using the hyperbolic plain waves, which are eigenfunctions of the hyperbolic Laplacian. Since the hyperbolic plane wave $\exp((\frac{n-1}{2}-i\lambda)\langle\langle\boldsymbol{\omega},\boldsymbol{x}\rangle_H)$ is an eigenfunction of the real operator $\mathcal{L}$ with real eigenvalue, so will this function multiplied by any phase $\exp(-ib)$, as will its real part. Call the result of this

$$\mathrm{HyLa}_{\lambda,b,\boldsymbol{\omega}}(\boldsymbol{z}) = \exp\left(\tfrac{n-1}{2}\langle\boldsymbol{\omega},\boldsymbol{z}\rangle_H\right)\cos\left(\lambda\langle\boldsymbol{\omega},\boldsymbol{z}\rangle_H + b\right). \tag{1}$$

This parameterization, which we call **HyLa** (for **Hy**perbolic **La**placian features), yields real-valued eigenfunctions of the Laplace-Beltrami operator with eigenvalue $-\lambda^2-(n-1)^2/4$. HyLa eigenfunctons have the nice property that they are bounded in almost every direction, as $\langle\boldsymbol{\omega},\boldsymbol{z}\rangle_H$ approaches 0 as $\boldsymbol{z}$ approaches any point on the boundary of $\mathcal{B}^n$ except $\boldsymbol{\omega}$. Note that HyLa eigenfunctions are invariant to isometries of the space: any isometric transformation of HyLa yields another HyLa eigenfunction with the same $\lambda$ but a transformed $\boldsymbol{\omega}$ (depending on how the isometry acts on the boundary $\partial\mathcal{B}^n$). It is easy to compute, parameterized by continuous instead of discrete parameters,

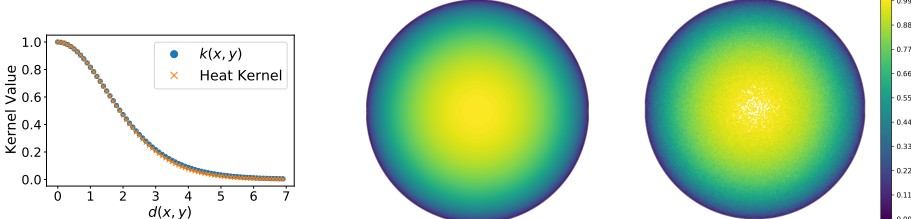

Figure 3: Visualization of the kernel $k(\boldsymbol{x}, \boldsymbol{y})$ when $\rho(\lambda) = \mathcal{N}(0, 0.5^2)$. (left) Distributions of $k(\boldsymbol{x}, \boldsymbol{y})$ and the heat kernel (at temperature $t = 6$) in 3D hyperbolic space; (middle) $k(\boldsymbol{x}, \boldsymbol{O})$ in 2D Poincaré disk model; (right) $\langle \phi(\boldsymbol{x}), \phi(\boldsymbol{y}) \rangle$ for HyLa features based on $D = 1000$ samples.

and are analogous to random Fourier features. Moreover, HyLa can be extended to eigenfunctions on other manifolds (e.g. symmetric spaces) since we only use manifold properties of $\mathbb{H}_n$.

We show that for any $\lambda \in \mathbb{R}$, under uniform sampling of $\boldsymbol{\omega}$ and $b$, the product $\mathrm{HyLa}_{\lambda,b,\boldsymbol{\omega}}(\boldsymbol{x}) \, \mathrm{HyLa}_{\lambda,b,\boldsymbol{\omega}}(\boldsymbol{y})$ is an unbiased estimate of an isometry-invariant kernel $k(\boldsymbol{x}, \boldsymbol{y})$.

**Theorem 4.1.** *Let $\boldsymbol{\omega}$ be sampled uniformly (under the Euclidean metric) from the boundary $\partial \mathcal{B}^n$ and let $b$ be sampled uniformly from $[0, 2\pi]$. Then $\mathbb{E}\left[\mathrm{HyLa}_{\lambda,b,\boldsymbol{\omega}}(\boldsymbol{x}) \, \mathrm{HyLa}_{\lambda,b,\boldsymbol{\omega}}(\boldsymbol{y})\right] = k_\lambda(\boldsymbol{x}, \boldsymbol{y})$ for any $\lambda \in \mathbb{R}$ and $\boldsymbol{x}, \boldsymbol{y} \in \mathbb{H}_n$, where the function $k_\lambda$ is an isometry-invariant kernel given by*

$$k_\lambda(\boldsymbol{x}, \boldsymbol{y}) = \tfrac{1}{2} \cdot {}_2F_1 \left( \tfrac{n-1}{2} + i\lambda, \tfrac{n-1}{2} - i\lambda; \tfrac{n}{2}; \tfrac{1}{2}\left(1 - \cosh(d_\mathbb{H}(\boldsymbol{x}, \boldsymbol{y}))\right) \right),$$

*where ${}_2F_1$ is the hypergeometric function, defined via analytic continuation by the power series*

$${}_2F_1\left(a, b; c; z\right) = \sum_{n=0}^{\infty} \frac{(a)_n (b)_n}{(c)_n} \frac{z^n}{n!} = 1 + \frac{ab}{c} \frac{z}{1!} + \frac{a(a+1)b(b+1)}{c(c+1)} \frac{z^2}{2!} + \cdots \qquad (|z| < 1).$$

Note that despite the presence of an $i$ in the formula, this kernel is clearly real because the hypergeometric function satisfies the properties ${}_2F_1(a, b; c; z) = {}_2F_1(b, a; c; z)$ and ${}_2F_1(a, b; c; z)^* = {}_2F_1(a^*, b^*; c^*; z^*)$. In practice, as with random Fourier features, instead of choosing one single $\lambda$, we select them at random from some distribution $\rho(\lambda)$. The resulting kernel will be an isometry-invariant kernel that depends on the distribution of $\lambda$, as follows:

$$k(\boldsymbol{x}, \boldsymbol{y}) = \tfrac{1}{2} \int_{-\infty}^{\infty} {}_2F_1 \left( \tfrac{n-1}{2} + i\lambda, \tfrac{n-1}{2} - i\lambda; \tfrac{n}{2}; \tfrac{1}{2}\left(1 - \cosh(d_\mathbb{H}(\boldsymbol{x}, \boldsymbol{y}))\right) \right) \cdot \rho(\lambda) \, d\lambda. \qquad (2)$$

This formula gives a way to derive the isometry-invariant kernel from a distribution $\rho(\lambda)$; if we are interested in finding a feature map for some particular kernel, we can invert this mapping to get a distribution for $\lambda$ which will produce the desired kernel.

**Theorem 4.2.** *Suppose that $k(\boldsymbol{x}, \boldsymbol{y}) = k(d_\mathbb{H}(\boldsymbol{x}, \boldsymbol{y}))$ is an isometry-invariant positive semidefinite kernel. Assume the existence of an associated density $\rho(\lambda)$ with the kernel, then*

$$\rho(\lambda) = \lambda \tanh\left(\tfrac{\pi\lambda}{2}\right) \int_{\mathcal{B}^n} \int_{\partial \mathcal{B}^n} k(d_\mathbb{H}(\boldsymbol{z}, \boldsymbol{O})) \exp\left( \left(\tfrac{n-1}{2} - i\lambda\right) \langle \boldsymbol{\omega}, \boldsymbol{z} \rangle_H \right) \, d\boldsymbol{\omega} \, d\boldsymbol{z}.$$

*i.e., $\rho(\lambda)$ is the spherical transform of the kernel, and if we draw $\lambda$ proportional to $\rho$, $\boldsymbol{\omega}$ uniformly on $\partial \mathcal{B}^n$, and $b$ uniformly on $[0, 2\pi]$, then*

$$k(0) \cdot \mathbb{E}\left[\mathrm{HyLa}_{\lambda,b,\boldsymbol{\omega}}(\boldsymbol{x}) \, \mathrm{HyLa}_{\lambda,b,\boldsymbol{\omega}}(\boldsymbol{y})\right] = k_\lambda(\boldsymbol{x}, \boldsymbol{y}) = k(\boldsymbol{x}, \boldsymbol{y}).$$

Although Theorem 4.2 lets us find a HyLa distribution for any isometric kernel, for simplicity in this paper, because the closed-forms of many kernels in $\mathbb{H}_n$ are not available, rather than arriving at a distribution via this inverse, we will instead focus on the case where $\rho$ is a Gaussian. This corresponds closely to a heat kernel (Grigor'yan & Noguchi, 1998), as illustrated in Figure 3 (left).

We will use HyLa eigenfunctions to produce Euclidean features from hyperbolic embeddings, using the same random-features approach as Rahimi et al. (2007). Concretely, to map from $\mathbb{H}_n$ to $\mathbb{R}^D$, we draw $D$ independent samples $\lambda_1, \ldots, \lambda_D$ from $\rho$, $D$ independent samples $\boldsymbol{\omega}_1, \ldots, \boldsymbol{\omega}_D$ uniform from $\partial \mathcal{B}^n$, and $D$ independent samples $b_1, \ldots, b_D$ uniform from $[0, 2\pi]$, and then output a feature map $\phi$ the $k$th coordinate of which is $\phi_k(\boldsymbol{x}) = \frac{1}{\sqrt{D}} \mathrm{HyLa}_{\lambda_k, b_k, \boldsymbol{\omega}_k}(\boldsymbol{x})$. It is easy to see that this will yield feature vectors with $\mathbb{E}\left[\langle \phi(\boldsymbol{x}), \phi(\boldsymbol{y}) \rangle\right] = k(\boldsymbol{x}, \boldsymbol{y})$ as given in Eq. 2.

We visualize the kernel $k(\boldsymbol{x}, \boldsymbol{O})$ for $\rho(\lambda) = \mathcal{N}(0, 0.25)$ in the 2-dimensional Poincarè disk in Figure 3, evaluating the integral in Eq. 2 using Gauss–Hermite quadrature. In Figure 3 (right), we sample random HyLa features with $D = 1000$ and plot $\langle \phi(\boldsymbol{x}), \phi(\boldsymbol{O}) \rangle$. Visibly, the HyLa features approximate the kernel well. A discussion of the estimation error is provided in Appendix B.

**Connection to Euclidean Activation.** There is a close connection between the HyLa eigenfunction and the Euclidean activations used in Euclidean fully connected networks. Given a data point $\boldsymbol{x} \in \mathbb{R}^n$, a weight $\boldsymbol{\omega} \in \mathbb{R}^n$, a bias $b \in \mathbb{R}$ and nonlinearity $\sigma$, a Euclidean DNN activation can be

written as $\sigma(\langle \boldsymbol{\omega}, \boldsymbol{x} \rangle + b) = \sigma(\|\boldsymbol{\omega}\| \langle \frac{\boldsymbol{\omega}}{\|\boldsymbol{\omega}\|}, \boldsymbol{x} \rangle + b)$. In hyperbolic space, for $\boldsymbol{z} \in \mathcal{B}^n$, $\boldsymbol{\omega} \in \partial \mathcal{B}^n$, $\lambda \in \mathbb{R}$, $b \in \mathbb{R}$ and $\sigma = \cos$, we can reformulate the HyLa eigenfunction as $\mathrm{HyLa}_{\lambda, b, \boldsymbol{\omega}}(\boldsymbol{z}) = \sigma\left(\lambda \langle \boldsymbol{\omega}, \boldsymbol{z} \rangle_H + b\right) \exp\left(\frac{n-1}{2} \langle \boldsymbol{\omega}, \boldsymbol{z} \rangle_H\right)$, HyLa generalizes Euclidean activations to hyperbolic space, with an extra factor $\exp\left(\frac{n-1}{2} \langle \boldsymbol{\omega}, \boldsymbol{z} \rangle_H\right)$ from the curvature of $\mathbb{H}_n$.

From a functional perspective, any $f \in L^2(\mathbb{H}_n)$ can be expanded as an infinite linear combination (integral form) of HyLa (Theorem 4.3 in Sonoda et al. (2022)). This statement holds whenever the non-linearity $\sigma$ is a tempered distribution on $\mathbb{R}$, i.e., the topological dual of the Schwartz test functions, including $\mathrm{ReLU}$ and $\cos$. Though the features will not approximate a kernel on hyperbolic space if $\sigma \neq \cos$, this suggests variants of HyLa with other nonlinearities may be interesting to study.

## 5 HyLa for Graph Learning

In this section, we show how to use HyLa to encode geometric priors for end-to-end graph learning.

**Background on Graph Learning.** A graph is defined formally as $\mathcal{G} = (\mathcal{V}, \mathbf{A})$, where $\mathcal{V}$ represents the vertex set consisting of $n$ nodes and $\mathbf{A} \in \mathbb{R}^{n \times n}$ represents the symmetric adjacency matrix. Besides the graph structure, each node in the graph has a corresponding $d$-dimensional feature vector: we let $\mathbf{X} \in \mathbb{R}^{n \times d}$ denote the entire feature matrix for all nodes. A fraction of nodes are associated with a label indicating one (or multiple) categories it belongs to. The *node classification* task is to predict the labels of nodes without labels or even of nodes newly added to the graph.

An important class of Euclidean graph learning model is the graph convolutional neural network (GCN) (Kipf & Welling, 2016; Defferrard et al., 2016). The GCN is widely used in graph tasks including semi-supervised learning for node classification, supervised learning for graph-level classification, and unsupervised learning for graph embedding. Many complex graph networks and GCN variants have been developed, such as the graph attention networks (GAT, Veličković et al. (2017)), FastGCN (Chen et al., 2018), GraphSage (Hamilton et al., 2017), and others (Velickovic et al., 2019; Xu et al., 2018). An interesting work to understand GCN is *simplifying GCN* (SGC, Wu et al. (2019)): a linear model derived by removing the non-linearities in a $K$-layer GCN as: $f(\mathbf{A}, \mathbf{X}) = \mathrm{softmax}(\mathbf{S}^K \mathbf{X} \mathbf{W})$, where $\mathbf{S}$ is the "normalized" adjacency matrix with added self-loops and $\mathbf{W}$ is the trainable weight. Note that the pre-processed features $\mathbf{S}^K \mathbf{X}$ can be computed before training, which enables large graph learning and greatly saves memory and computation cost.

**End-to-End Learning with HyLa.** We propose a feature-extracted architecture via the following recipe: embed the data objects (graph nodes or features, detailed below) into some space (e.g., Euclidean or hyperbolic), map the embedding to Euclidean features $\overline{\mathbf{X}}$ via the kernel transformation (e.g., RFF or HyLa), and finally apply an Euclidean graph learning model $f(\mathbf{A}, \overline{\mathbf{X}})$. This recipe only manipulates the input of the graph learning model and hence, this architecture can be used with *any* graph learning model. The graph model and the hyperbolic embedding are learned simultaneously with backpropagation. In theory, the embedding space can be any desired space, just use the same Laplacian recipe to construct features. Below we only show the pipeline for hyperbolic space, while we also include Euclidean space (with random Fourier features) as a baseline in our experiments.

**Directly Embed Graph Nodes.** We embed each node into a low dimensional hyperbolic space $\mathcal{B}^{d_0}$ as hyperbolic embedding $\mathbf{Z} \in \mathbb{R}^{n \times d_0}$ for all nodes in the graph, which can be either a pretrained fixed embedding or as parameters learnt together with the subsequent graph learning model during training time. To compute with the HyLa eigenfunctions, first sample $d_1$ points uniformly from the boundary $\partial \mathcal{B}^{d_0}$ to get $\boldsymbol{\Omega} \in \mathbb{R}^{d_1 \times d_0}$, then sample $d_1$ eigenvalues and biases separately from

---

**Algorithm 1** End-to-End HyLa

**input:** $n$ objects, Poincaré disk $\mathcal{B}^{d_0}$, HyLa feature dimension $d_1$, adjacency matrix $\mathbf{A}$, node feature matrix $\mathbf{X}$, graph neural network $f$
**initialize** $\mathbf{Z} \in \mathbb{R}^{n \times d_0}$ {hyperbolic embeddings}
**sample** boundary pts matrix $\boldsymbol{\Omega} \in \mathbb{R}^{d_1 \times d_0}$, eigenvalues $\boldsymbol{\Lambda} \in \mathbb{R}^{n \times d_1}$ and biases $\mathbf{B} \in \mathbb{R}^{n \times d_1}$
**compute** $\mathbf{P} = \langle \boldsymbol{\Omega}, \mathbf{Z} \rangle_H \in \mathbb{R}^{n \times d_1}$ {Horocycle distance}
**compute** $\overline{\mathbf{X}} = \exp\left(\frac{n-1}{2} \mathbf{P}\right) \cos(\boldsymbol{\Lambda} \cdot \mathbf{P} + \mathbf{B})$ {HyLa}
**if embedding features:** $\overline{\mathbf{X}} = \mathbf{X} \overline{\mathbf{X}} \in \mathbb{R}^{n \times d_1}$
**return** $\mathbf{Y} = f(\mathbf{A}, \overline{\mathbf{X}})$ {e.g., SGC}

---

$\mathcal{N}(0, s^2)$ and $\mathrm{Uniform}([0, 2\pi])$ to get $\boldsymbol{\Lambda}, \mathbf{B} \in \mathbb{R}^{d_1}$, where $s$ is a scale constant. The resulting feature matrix ($\overline{\mathbf{X}} \in \mathbb{R}^{n \times d_1}$) computation follows; please refer to Algorithm 1 for a detailed breakdown. The mapped features $\overline{\mathbf{X}}$ are then fed into the chosen graph learning model $f(\mathbf{A}, \overline{\mathbf{X}})$ for prediction.

**Embed Features.** One can also embed the given features into hyperbolic space so as to derive the node features implicitly. Specifically, when a node feature matrix $\mathbf{X} \in \mathbb{R}^{n \times d}$ is given, we initialize a hyperbolic embedding for each of the $d$ dimensions to derive hyperbolic embeddings $\mathbf{Z} \in \mathbb{R}^{d \times d_0}$. The Euclidean features $\overline{\mathbf{X}}$ can be computed in the same manner following Algorithm 1. However, to get the new node feature matrix, an extra aggregation step $\overline{\mathbf{X}} = \mathbf{X}\overline{\mathbf{X}} \in \mathbb{R}^{n \times d_1}$ is required before fedding into a graph learning model.

Embedding graph nodes is better-suited for tasks where no feature matrix is available or meaningful features are hard to derive. However, the size of the embedding $\mathbf{Z}$ will be proportional to the graph size, hence it may not scale to very large graphs due to memory and computation constraints. Furthermore, this method can only be used in a transductive setting, where nodes in the test set are seen during training. In comparison, embedding features can be used even for large graphs since the dimension $d$ of the original feature matrix is usually fixed and much lower than the number of nodes $n$. Note that as the hyperbolic embeddings are built for each feature dimension, they can be used in both transductive and inductive settings, as long as the test data shares the same set of features as the training data. One limitation is that its performance depends on the existence of a feature matrix $\mathbf{X}$ that contains sufficient information for learning.

*Any* graph learning model can be used in the proposed feature-extracted architecture. In our experiments, we focus primarily on the simple linear graph network SGC, which takes the form $f(\mathbf{A}, \overline{\mathbf{X}}) = \mathrm{softmax}(\mathbf{A}^K \overline{\mathbf{X}} \mathbf{W})$ with a trainable weight matrix $\mathbf{W}$. Note that just as the vanilla SGC case, $\mathbf{A}^K$ or $\mathbf{A}^K \mathbf{X}$ can be pre-computed in the same way before training and inference. For the purpose of end-to-end learning, we jointly learn the embedding parameter $\mathbf{Z}$ and weight $\mathbf{W}$ in SGC during the training time. It's also possible to adopt a two-step approach, i.e., first pretrain a hyperbolic embedding following (Nickel & Kiela, 2017; 2018; Sala et al., 2018; Sonthalia & Gilbert, 2020; Chami et al., 2019), then fix the embedding and train the graph learning model only. We defer this discussion to Appendix D due to page limit.

## 6 Experiments

### 6.1 Node Classification

**Task and Datasets.** The goal of this task is to classify each node into a correct category. We use transductive datasets: Cora, Citeseer and Pubmed (Sen et al., 2008), which are standard citation networks benchmarks, following the standard splits adopted in Kipf & Welling (2016). We also include datasets adopted in HGCN (Chami et al., 2019) for comparison: disease propagation tree and Airport. Yhe former contains tree networks simulating the SIR disease spreading model (Anderson & May, 1992), while the latter contains airline routes between airports from OpenFlights. To measure scalability, we supplement our experiment by predicting community structure on a large inductive dataset Reddit following Wu et al. (2019). More experimental details are provided in Appendix E.

**Experiment Setup.** Since all datasets contain node features, we choose to embed features most of the time, since it applies to both small and large graphs, and transductive and inductive tasks. The only exception is the Airport dataset, which contains only 4 dimensional features—here, we use HyLa/RFF after embedding the graph nodes to produce better features $\overline{\mathbf{X}}$. We then use SGC model as $\mathrm{softmax}(\mathbf{A}^K \overline{\mathbf{X}} \mathbf{W})$, where both $\mathbf{W}$ and $\mathbf{Z}$ are jointly learned.

**Baselines.** On Disease, Airport, Pubmed, Citeseer and Cora dataset, we compare our HyLa/RFF-SGC model against both Euclidean models (GCN, SGC and GAT) and Hypebolic models (HGCN, LGCN and HNN) using their publicly released version in Table 1, where all hyperbolic models adopt a *16-dimensional* hyperbolic space for consistency and a fair comparison. For the largest Reddit dataset, a 50-dimensional hyperbolic space is used. We also compare against the reported performance of supervised and unsupervised variants of GraphSAGE and FastGCN in Table 1. Note that GCN-based models (e.g., HGCN, LGCN) could not be trained on Reddit because its adjacency matrix is too large to fit in memory, unless a sampling way is used for training. Worthy to mention, despite of the fact that standard Euclidean GCN literature (Kipf & Welling, 2016; Wu et al., 2019) train the model for *100* epochs on the node classification task, most hyperbolic (graph) networks including HGCN, LGCN, GIL[1] report results of training for (5-)thousand epochs with early stopping.

---

[1]We cannot replicate results of GIL from their public code.

Table 1: Test accuracy/Micro F1 Score (%) averaged over 10 runs on node classification task. Performance of some baselines are taken from their original papers. **OOM:** Out of memory.

| Dataset | Disease | Airport | Pubmed | Citeseer | Cora | | Model | Test F1 |
|---|---|---|---|---|---|---|---|---|
| Hyperbolicity $\delta$ | 0 | 1.0 | 3.5 | 5.0 | 11 | | GCN | **OOM** |
| GCN | $69.7 \pm 0.4$ | $81.4 \pm 0.6$ | $78.1 \pm 0.2$ | $70.5 \pm 0.8$ | $81.3 \pm 0.3$ | | HGCN | **OOM** |
| SGC | $69.5 \pm 0.2$ | $80.6 \pm 0.1$ | $78.9 \pm 0.0$ | $71.9 \pm 0.1$ | $81.0 \pm 0.0$ | | LGCN | **OOM** |
| GAT | $70.4 \pm 0.4$ | $81.5 \pm 0.3$ | $79.0 \pm 0.3$ | $72.5 \pm 0.7$ | $\mathbf{83.0} \pm 0.7$ | | SAGE-mean | 95.0 |
| HGCN | $74.5 \pm 0.9$ | $90.6 \pm 0.2$ | $\mathbf{80.3} \pm 0.3$ | $64.0 \pm 0.6$ | $79.9 \pm 0.2$ | | SAGE-GCN | 93.0 |
| LGCN | $81.3 \pm 4.0$ | $57.6 \pm 0.7$ | $77.8 \pm 0.7$ | $65.9 \pm 0.8$ | $78.0 \pm 0.6$ | | FastGCN | 93.7 |
| HNN | $41.0 \pm 1.8$ | $80.5 \pm 0.5$ | $69.8 \pm 0.4$ | $52.0 \pm 1.0$ | $54.6 \pm 0.4$ | | SGC | 94.9 |
| RFF-SGC | $83.4 \pm 1.9$ | $94.8 \pm 0.8$ | $77.6 \pm 0.1$ | $71.3 \pm 0.6$ | $81.6 \pm 0.4$ | | RFF-SGC | 93.9 |
| HyLa-SGC | $\mathbf{86.8} \pm 2.1$ | $\mathbf{95.2} \pm 0.5$ | $\mathbf{80.3} \pm 0.9$ | $\mathbf{72.6} \pm 1.0$ | $82.5 \pm 0.5$ | | HyLa-SGC | 94.5 |

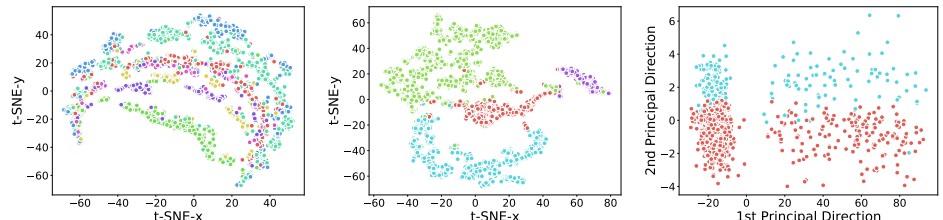

Figure 4: Visualization of node HyLa features on Cora, Airport and Disease datasets, where nodes of different classes are indicated by different colors. (left) t-SNE on Cora; (middle) t-SNE on Airport; (right) PCA on Disease.

For a fair comparison, we train all models for a maximum of *100* epochs with early stopping, except from HGCN[2], whose results were taken from the original paper trained for 5,000 epochs.

**Analysis.** Our feature-extracted architecture is particularly strong and expressive to encode geometric priors for graph learning from Table 1. Together with SGC, HyLa-SGC outperforms state-of-the-art hyperbolic models on nearly all datasets. In particular, HyLa-SGC beats not only Euclidean SGC/GCN, but also an attention model GAT, except on the Cora dataset with a comparable performance. For the tree network disease with lowest hyperbolicity (more hyperbolic), the improvements of HyLa-SGC over HGCN is about 12.3%! The results suggest that the more hyperbolic the graph is, the more improvements will be gained with HyLa. On Reddit, HyLa-SGC outperforms sampling-based GCN variants, SAGE-GCN and FastGCN by more than 1%. However, the performance is close to SGC, which may indicate that the extra weights and nonlinearities are unnecessary for this particular dataset. Notably, RFF-SGC, embedding into Euclidean space and using RFF, sometimes can be better than GCN/SGC, while HyLa-SGC is consistently bettter than RFF-SGC.

**Visualization.** In Figure 4, we visualize the learned node Euclidean features using HyLa on Cora, Airport and Disease datasets with t-SNE (Van der Maaten & Hinton, 2008) and PCA projection. This shows that HyLa achieves great label class separation (indicated by different colors).

## 6.2 TEXT CLASSIFICATION

**Task and Datsets.** We further evaluate HyLa on **transductive** and **inductive** text classification task to assign documents labels. We conducted experiments on 4 standard benchmarks including R52 and R8 of Reuters 21578 dataset, Ohsumed and Movie Review (MR) follows the same data split as Yao et al. (2019); Wu et al. (2019). Detailed dataset statistics are provided in Table 4.

**Experiment Setup.** In the transductive case, previous work Yao et al. (2019) and Wu et al. (2019) apply GCN and SGC by creating a corpus-level graph where both documents and words are treated as nodes in the graph. For weights and connections in the graph, word-word edge weights are calculated as pointwise mutual information (PMI) and word-document edge weights as normalized TF-IDF scores. The weights of document-document edges are unknown and left as 0. We follows the same data processing setup for the transductive setting, and embed the whole graph since only the adjacent matrix is available. In the inductive setting, we take the sub-matrix of the large matrix in

---

[2]HGCN requires pretraining embeddings from a link prediction task to achieve reported results on node classification task for Pubmed and Cora.

Table 2: Test accuracy (%) averaged over 10 runs on transductive and inductive text classification task except from the LR mode. **Bold** numbers: best in both transductive and inductive setting; Underlined numbers: best in inductive setting.

| Setting | Methods | R8 | R52 | Ohsumed | MR |
|---|---|---|---|---|---|
| Trans-ductive | TextGCN | $97.1 \pm 0.1$ | $93.5 \pm 0.2$ | $68.4 \pm 0.6$ | $\mathbf{76.7} \pm 0.2$ |
| | TextSGC | $97.2 \pm 0.1$ | $94.0 \pm 0.2$ | $\mathbf{68.5} \pm 0.3$ | $75.9 \pm 0.3$ |
| | RFF-SGC | $96.5 \pm 0.3$ | $94.0 \pm 0.5$ | $67.2 \pm 0.4$ | $73.1 \pm 0.4$ |
| | HyLa-SGC | $96.9 \pm 0.4$ | $\mathbf{94.1} \pm 0.3$ | $67.3 \pm 0.5$ | $76.2 \pm 0.3$ |
| Inductive | TextGCN | $95.8 \pm 0.3$ | $88.2 \pm 0.7$ | $57.7 \pm 0.4$ | $74.8 \pm 0.3$ |
| | LR | $93.3$ | $85.6$ | $56.6$ | $73.0$ |
| | HyLa-LR | $\underline{\mathbf{97.4}} \pm 0.2$ | $\underline{93.5} \pm 0.2$ | $\underline{64.9} \pm 0.3$ | $\underline{75.5} \pm 0.3$ |
| | RFF-LR | $97.0 \pm 0.4$ | $92.2 \pm 0.2$ | $61.6 \pm 0.3$ | $\mathbf{76.0} \pm 0.3$ |

the transductive setting, including only the document-word edges as the node representation feature matrix $\mathbf{X}$, then follow the procedure in Section 5 to embed features and apply HyLa/RFF to get $\overline{\mathbf{X}}$. Since the adjacency matrix of documents is unknown, we replace SGC with a logistic regression (LR) formalized as $\mathbf{Y} = \text{softmax}(\overline{\mathbf{X}}\mathbf{W})$. We train all models for a maximum of 200 epochs and compare it against TextSGC and TextGCN in Table 2.

**Performance Analysis.** In the transductive setting, HyLa-based models can match the performance of TextGCN and TextSGC. The corpus-level graph may contain sufficient information to learn the task, and hence HyLa-SGC does not seem to outperform baselines, but still has a comparable performance. HyLa shows extraordinary performance in the inductive setting, where less information is used compared to the (transductive) node level case, i.e. a submatrix of corpus-level graph. With a linear regression model, it can already outperform inductive TextGCN, sometimes even better than the performance of a transductive TextGCN model, which indicates that there is indeed redundant information in the corpus-level graph. From the results on the inductive text classification task, we argue that HyLa (with features embedding) is particularly useful in the following three ways. First, it can solve the OOM problem of classic GCN model in large graphs, and requires less memory during training, since there are limited number of lower level features (also $\mathbf{X}$ is usually sparse), and there is no need to build a corpus-level graph anymore. Second, it is naturally inductive as HyLa is built at feature level (for each word in this task), it generalizes to any unseen new nodes (documents) that uses the same set of words. Third, the model is simple: HyLa follows by a linear regression model, which computes faster than classical GCN models.

**Efficiency.** Following Wu et al. (2019), we measure the training time of HyLa-based models on the Pubmed dataset, and we compare against both Euclidean models (SGC, GCN, GAT) and Hyperbolic models (HGCN, HNN). In Figure 5, we plot the timing performance of various models, taking into account the pre-computation time of the models into training time. We measure the training time on a NVIDIA GeForce RTX 2080 Ti GPU and show the specific timing statistics in Appendix. HyLa-based models achieve the best performance while incurring a minor computational slowdown, which is $4.4\times$ faster than HGCN.

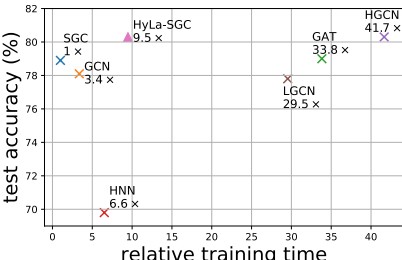

Figure 5: Performance over training time on Pubmed. HyLa-SGC achieves best performance with minor computation slowdown.

## 7 CONCLUSION

We propose a simple and efficient approach to using hyperbolic space in neural networks, by deriving HyLa as an expressive feature using the Laplacian eigenfunctions. Empirical results on graph learning tasks show that HyLa can outperform SOTA hyperbolic networks. HyLa sheds light as a principled approach to utilizing hyperbolic geometry in an entirely different way to previous work. Possible future directions include (1) using HyLa with non-linear graph networks such as GCN to derive even more expressive models; and (2) adopting more numerically stable representations of the hyperbolic embeddings to avoid potential "NaN problems" when learning with HyLa.

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

## A  THEOREMS

**Theorem A.1** (Helgason (2022); Zelditch (2017)). *All smooth eigenfunctions of the Euclidean Laplacian $\Delta$ on $\mathbb{R}^n$ are*

$$f(\boldsymbol{x}) = \int_{S^{n-1}} e^{i\lambda\langle\boldsymbol{\omega},\boldsymbol{x}\rangle} dT(\boldsymbol{\omega}),$$

*where $\lambda \in \mathbb{C} - \{0\}$ and $T$ is an analytic functional (or hyperfunction), i.e., an element of the dual space of the space of analytic functions on $S^{n-1}$.*

**Theorem A.2** (Helgason (2022); Zelditch (2017)). *All smooth eigenfunctions of the Hyperbolic Laplacian $\mathcal{L}$ on $\mathcal{B}^n$ are*

$$f(\boldsymbol{z}) = \int_{\partial\mathcal{B}^n} \exp((i\lambda + \tfrac{n-1}{2})\langle\boldsymbol{\omega},\boldsymbol{z}\rangle_H)\, dT(\boldsymbol{\omega}),$$

*where $\lambda \in \mathbb{C}$ and $T$ is an analytic functional (or hyperfunction).*

**Lemma A.3** (Expression of $k(\boldsymbol{x},\boldsymbol{O})$). *Denote $\zeta_{\lambda,\boldsymbol{\omega}}(\boldsymbol{z}) = \exp\left((\tfrac{n-1}{2} + i\lambda)\langle\boldsymbol{\omega},\boldsymbol{z}\rangle_H\right)$, then the corresponding kernel $k_\lambda(\boldsymbol{x},\boldsymbol{O})$ for a particular value of $\lambda$ defined as*

$$k_\lambda(\boldsymbol{x},\boldsymbol{O}) = \mathbb{E}\left[\mathrm{HyLa}_{\lambda,b,\boldsymbol{\omega}}(\boldsymbol{x})\,\mathrm{HyLa}_{\lambda,b,\boldsymbol{\omega}}(\boldsymbol{y})\right] = \frac{1}{2}\cdot\mathbb{E}_{\boldsymbol{\omega}}\left[\zeta_{\lambda,\boldsymbol{\omega}}(\boldsymbol{O})^*\zeta_{\lambda,\boldsymbol{\omega}}(\boldsymbol{x})\right] = \frac{1}{2}\cdot\mathbb{E}_{\boldsymbol{\omega}}\left[\zeta_{\lambda,\boldsymbol{\omega}}(\boldsymbol{x})\right] \quad (3)$$

*takes the form*

$$k_\lambda(\boldsymbol{x},\boldsymbol{O}) = \frac{1}{2}\cdot {}_2F_1\left(\frac{n-1}{2} + i\lambda, \frac{n-1}{2} - i\lambda; \frac{n}{2}; \frac{1}{2}\left(1 - \cosh(d_{\mathbb{H}}(\boldsymbol{x},\boldsymbol{O}))\right)\right),$$

*where the expectation in Equation 3 is taken over $\boldsymbol{\omega}$ drawn uniformly from the $n$-dimensional unit sphere.*

*Proof.* Recall that for $\boldsymbol{x}$ in the $n$-dimensional Poincare ball $\mathcal{B}^n$ and $\boldsymbol{\omega} \in \partial\mathcal{B}^n$,

$$\langle\boldsymbol{\omega},\boldsymbol{x}\rangle_H = \log\left(\frac{1 - \|\boldsymbol{x}\|^2}{\|\boldsymbol{x} - \boldsymbol{\omega}\|^2}\right).$$

Let $\gamma = \frac{n-1}{2} + i\lambda$, expanding Equation 3 out,

$$\begin{aligned}
k_\lambda(\boldsymbol{x},\boldsymbol{O}) &= \frac{1}{2}\cdot\mathbb{E}_{\boldsymbol{\omega}}\left[\exp\left(\gamma\log\left(\frac{1 - \|\boldsymbol{x}\|^2}{\|\boldsymbol{x} - \boldsymbol{\omega}\|^2}\right)\right)\right]\\
&= \frac{1}{2}\cdot\exp\left(\gamma\log\left(\frac{1 - \|\boldsymbol{x}\|^2}{1 + \|\boldsymbol{x}\|^2}\right)\right)\mathbb{E}_{\boldsymbol{\omega}}\left[\exp\left(\gamma\log\left(\frac{1 + \|\boldsymbol{x}\|^2}{\|\boldsymbol{x} - \boldsymbol{\omega}\|^2}\right)\right)\right]\\
&= \frac{1}{2}\cdot\exp\left(\gamma\log\left(\frac{1 - \|\boldsymbol{x}\|^2}{1 + \|\boldsymbol{x}\|^2}\right)\right)\mathbb{E}_{\boldsymbol{\omega}}\left[\left(\frac{\|\boldsymbol{x} - \boldsymbol{\omega}\|^2}{1 + \|\boldsymbol{x}\|^2}\right)^{-\gamma}\right]\\
&= \frac{1}{2}\cdot\exp\left(\gamma\log\left(\frac{1 - \|\boldsymbol{x}\|^2}{1 + \|\boldsymbol{x}\|^2}\right)\right)\mathbb{E}_{\boldsymbol{\omega}}\left[\left(\frac{\|\boldsymbol{x}\|^2 - 2\boldsymbol{\omega}^T\boldsymbol{x} + \|\boldsymbol{\omega}\|^2}{1 + \|\boldsymbol{x}\|^2}\right)^{-\gamma}\right],
\end{aligned}$$

and if we let

$$u = \frac{-2\|\boldsymbol{x}\|}{1 + \|\boldsymbol{x}\|^2},$$

and let $\hat{\boldsymbol{x}}$ denote the unit vector in the direction of $\boldsymbol{x}$, then this becomes

$$k_\lambda(\boldsymbol{x},\boldsymbol{O}) = \frac{1}{2}\cdot\exp\left(\gamma\log\left(\frac{1 - \|\boldsymbol{x}\|^2}{1 + \|\boldsymbol{x}\|^2}\right)\right)\mathbb{E}_{\boldsymbol{\omega}}\left[\left(1 + u\hat{\boldsymbol{x}}^T\boldsymbol{\omega}\right)^{-\gamma}\right].$$

Expanding this out using the Binomial Formula (which is valid here because $|u| < 1$), we get

$$\begin{aligned}
k_\lambda(\boldsymbol{x},\boldsymbol{O}) &= \frac{1}{2}\cdot\exp\left(\gamma\log\left(\frac{1 - \|\boldsymbol{x}\|^2}{1 + \|\boldsymbol{x}\|^2}\right)\right)\mathbb{E}_{\boldsymbol{\omega}}\left[\sum_{k=0}^{\infty}\binom{-\gamma}{k}u^k\left(\hat{\boldsymbol{x}}^T\boldsymbol{\omega}\right)^k\right]\\
&= \frac{1}{2}\cdot\exp\left(\gamma\log\left(\frac{1 - \|\boldsymbol{x}\|^2}{1 + \|\boldsymbol{x}\|^2}\right)\right)\sum_{k=0}^{\infty}\binom{-\gamma}{k}u^k\mathbb{E}_{\boldsymbol{\omega}}\left[\left(\hat{\boldsymbol{x}}^T\boldsymbol{\omega}\right)^k\right].
\end{aligned}$$

Since this expected value is $0$ for odd $k$, this becomes

$$k_\lambda(\boldsymbol{x}, \boldsymbol{O}) = \frac{1}{2} \cdot \exp\left(\gamma \log\left(\frac{1 - \|\boldsymbol{x}\|^2}{1 + \|\boldsymbol{x}\|^2}\right)\right) \sum_{k=0}^{\infty} \binom{-\gamma}{2k} u^{2k} \operatorname*{\mathbb{E}}_{\boldsymbol{\omega}}\left[\left(\hat{\boldsymbol{x}}^T \boldsymbol{\omega}\right)^{2k}\right].$$

But $\hat{\boldsymbol{x}}^T \boldsymbol{\omega}$ has the same the distribution as the inner product of two uniform random unit vectors in $n$ dimensions. The square of this is well known to be Beta-distributed with parameters $(\frac{1}{2}, \frac{n-1}{2})$. So we can write this as

$$k_\lambda(\boldsymbol{x}, \boldsymbol{O}) = \frac{1}{2} \cdot \exp\left(\gamma \log\left(\frac{1 - \|\boldsymbol{x}\|^2}{1 + \|\boldsymbol{x}\|^2}\right)\right) \sum_{k=0}^{\infty} \binom{-\gamma}{2k} u^{2k} \operatorname*{\mathbb{E}}_{w}\left[w^k\right],$$

where $w \sim \mathrm{Beta}(\frac{1}{2}, \frac{n-1}{2})$. Of course, the moments of the Beta distribution are well-known to be

$$\operatorname*{\mathbb{E}}_{w}\left[w^k\right] = \frac{\left(\frac{1}{2}\right)^{(k)}}{\left(\frac{n}{2}\right)^{(k)}},$$

where $x^{(k)}$ denotes the Pochhammer symbol representing the rising factorial. On the other hand,

$$\binom{-\gamma}{2k} = \frac{(-\gamma)_{(2k)}}{(2k)!}$$

where $x_{(k)}$ denotes the Pochhammer symbol representing the falling factorial. Since $x_{(k)} = (-1)^k (-x)_{(k)}$, we can write this in terms of rising factorials as

$$\binom{-\gamma}{2k} = \frac{(\gamma)^{(2k)}}{(2k)!}.$$

So substituting everything in, we have

$$k_\lambda(\boldsymbol{x}, \boldsymbol{O}) = \frac{1}{2} \cdot \exp\left(\gamma \log\left(\frac{1 - \|\boldsymbol{x}\|^2}{1 + \|\boldsymbol{x}\|^2}\right)\right) \sum_{k=0}^{\infty} \frac{(\gamma)^{(2k)}}{(2k)!} \cdot u^{2k} \cdot \frac{\left(\frac{1}{2}\right)^{(k)}}{\left(\frac{n}{2}\right)^{(k)}}.$$

Next, observe that

$$\left(\frac{1}{2}\right)^{(k)} = \prod_{m=0}^{k-1}\left(\frac{1}{2} + m\right) = 2^{-k} \prod_{m=0}^{k-1}(2m + 1) = 4^{-k} \cdot \frac{(2k)!}{k!}.$$

So this becomes

$$k_\lambda(\boldsymbol{x}, \boldsymbol{O}) = \frac{1}{2} \cdot \exp\left(\gamma \log\left(\frac{1 - \|\boldsymbol{x}\|^2}{1 + \|\boldsymbol{x}\|^2}\right)\right) \sum_{k=0}^{\infty} \frac{(\gamma)^{(2k)}}{k!} \cdot u^{2k} \cdot 4^{-k} \cdot \frac{1}{\left(\frac{n}{2}\right)^{(k)}}.$$

Next, we leverage the famous identity that

$$\gamma^{(2k)} = 4^k \left(\frac{\gamma}{2}\right)^{(k)} \left(\frac{\gamma + 1}{2}\right)^{(k)}$$

to get

$$k_\lambda(\boldsymbol{x}, \boldsymbol{O}) = \frac{1}{2} \cdot \exp\left(\gamma \log\left(\frac{1 - \|\boldsymbol{x}\|^2}{1 + \|\boldsymbol{x}\|^2}\right)\right) \sum_{k=0}^{\infty} \frac{1}{k!} \cdot \left(\frac{\gamma}{2}\right)^{(k)} \cdot \left(\frac{\gamma + 1}{2}\right)^{(k)} \cdot u^{2k} \cdot \frac{1}{\left(\frac{n}{2}\right)^{(k)}}$$

$$= \frac{1}{2} \cdot \exp\left(\gamma \log\left(\frac{1 - \|\boldsymbol{x}\|^2}{1 + \|\boldsymbol{x}\|^2}\right)\right) \cdot {}_2F_1\left(\frac{\gamma + 1}{2}; \frac{\gamma}{2}; \frac{n}{2}; u^2\right).$$

Since

$$u^2 = \frac{4 \|\boldsymbol{x}\|^2}{1 + 2 \|\boldsymbol{x}\|^2 + \|\boldsymbol{x}\|^4},$$

it follows that

$$1 - u^2 = \frac{1 - 2\|\boldsymbol{x}\|^2 + \|\boldsymbol{x}\|^4}{1 + 2\|\boldsymbol{x}\|^2 + \|\boldsymbol{x}\|^4} = \left(\frac{1 - \|\boldsymbol{x}\|^2}{1 + \|\boldsymbol{x}\|^2}\right)^2$$

and

$$\frac{\sqrt{1 - u^2} - 1}{2\sqrt{1 - u^2}} = \frac{-\|\boldsymbol{x}\|^2}{1 - \|\boldsymbol{x}\|^2}.$$

Recall the classic formula (https://functions.wolfram.com/HypergeometricFunctions/Hypergeometric2F1/17/ShowAll.html) that

$$_2F_1\left(a, a + \frac{1}{2}; c; z\right) = (1 - z)^{-a}\,_2F_1\left(2a, 2c - 2a - 1; c; \frac{\sqrt{1 - z} - 1}{2\sqrt{1 - z}}\right).$$

Substituting $z = u^2$, $a = \frac{\gamma}{2}$, and $c = \frac{n}{2}$ yields

$$k_\lambda(\boldsymbol{x}, \boldsymbol{O}) = \frac{1}{2} \cdot {_2F_1}\left(\gamma, n - \gamma - 1; \frac{n}{2}; \frac{-\|\boldsymbol{x}\|^2}{1 - \|\boldsymbol{x}\|^2}\right)$$

$$= \frac{1}{2} \cdot {_2F_1}\left(\frac{n - 1}{2} + i\lambda, \frac{n - 1}{2} - i\lambda; \frac{n}{2}; \frac{-\|\boldsymbol{x}\|^2}{1 - \|\boldsymbol{x}\|^2}\right).$$

Therefore,

$$\frac{-\|\boldsymbol{x}\|^2}{1 - \|\boldsymbol{x}\|^2} = \frac{-\tanh^2(D/2)}{1 - \tanh^2(D/2)} = \frac{-\tanh^2(D/2)}{\text{sech}^2(D/2)} = -\sinh^2(D/2) = \frac{1}{2}(1 - \cosh(D)),$$

where $D = d_{\mathbb{H}}(\boldsymbol{x}, \boldsymbol{O})$. So we get a final expression

$$k_\lambda(\boldsymbol{x}, \boldsymbol{O}) = \frac{1}{2} \cdot {_2F_1}\left(\frac{n - 1}{2} + i\lambda, \frac{n - 1}{2} - i\lambda; \frac{n}{2}; \frac{1}{2}(1 - \cosh(d_{\mathbb{H}}(\boldsymbol{x}, \boldsymbol{O})))\right).$$

The proof is done here. We further make a remark here. Recall that for the Poincaré ball model,

$$d(\boldsymbol{x}, \boldsymbol{O}) = 2\log\left(\frac{\|\boldsymbol{x}\| + 1}{\sqrt{1 - \|\boldsymbol{x}\|^2}}\right)$$

$$= \log\left(\frac{(\|\boldsymbol{x}\| + 1)^2}{1 - \|\boldsymbol{x}\|^2}\right)$$

$$= \log\left(\frac{1 + \|\boldsymbol{x}\|}{1 - \|\boldsymbol{x}\|}\right)$$

$$= 2\,\text{artanh}(\|\boldsymbol{x}\|).$$

Therefore,

$$\log\left(\frac{1 - \|\boldsymbol{x}\|^2}{1 + \|\boldsymbol{x}\|^2}\right) = \log\left(\frac{1 - \tanh^2(D/2)}{1 + \tanh^2(D/2)}\right)$$

$$= \log\left(\frac{\text{sech}^2(D/2)}{1 + \tanh^2(D/2)}\right)$$

$$= \log\left(\frac{1}{\sinh^2(D/2) + \cosh^2(D/2)}\right)$$

$$= \log\left(\frac{1}{\sinh^2(D/2) + \cosh^2(D/2)}\right)$$

$$= -\log(\cosh(D)).$$

And on the other hand,

$$u = \frac{-2\tanh(D/2)}{1 + \tanh^2(D/2)} = \tanh(D).$$

then we can also write the kernel as

$$k_\lambda(\boldsymbol{x}, \boldsymbol{O}) = \frac{1}{2} \cdot \exp\left(-\left(\frac{n-1}{2} + i\lambda\right)\log\left(\cosh(D)\right)\right) \cdot {}_2F_1\left(\frac{n+1}{4} + i\frac{\lambda}{2}, \frac{n-1}{4} + i\frac{\lambda}{2}; \frac{n}{2}; \tanh(D)^2\right),$$

where $D = d_{\mathbb{H}}(\boldsymbol{x}, \boldsymbol{O})$. □

**Lemma A.4** (Isometry-invariance). *The kernel defined by*

$$k_\lambda(\boldsymbol{x}, \boldsymbol{y}) = \frac{1}{2} \cdot \underset{\boldsymbol{\omega}}{\mathbb{E}}\left[\zeta_{\lambda,\boldsymbol{\omega}}(\boldsymbol{x})^*\zeta_{\lambda,\boldsymbol{\omega}}(\boldsymbol{y})\right] = \frac{1}{2} \cdot \underset{\boldsymbol{\omega}}{\mathbb{E}}\left[\exp\left((\frac{n-1}{2} - i\lambda)\langle\boldsymbol{x}, \boldsymbol{\omega}\rangle_H + (\frac{n-1}{2} + i\lambda)\langle\boldsymbol{y}, \boldsymbol{\omega}\rangle_H\right)\right]$$

*is isometry-invariant.*

*Proof.* Let $g$ be any isometry of the space, then observe the geometric identity (Helgason, 2022):

$$\langle g \circ \boldsymbol{x}, g \circ \boldsymbol{\omega}\rangle = \langle\boldsymbol{x}, \boldsymbol{\omega}\rangle + \langle g \circ \boldsymbol{O}, g \circ \boldsymbol{\omega}\rangle. \tag{4}$$

Take $\boldsymbol{\omega} = g^{-1} \circ \boldsymbol{\omega}$ in Equation 4, it follows that

$$\langle g \circ \boldsymbol{x}, \boldsymbol{\omega}\rangle = \langle\boldsymbol{x}, g^{-1} \circ \boldsymbol{\omega}\rangle + \langle g \circ \boldsymbol{O}, \boldsymbol{\omega}\rangle.$$

Take $\boldsymbol{x} = g^{-1} \circ \boldsymbol{O}$ in Equation 4, it follows that

$$0 = \langle\boldsymbol{O}, \boldsymbol{\omega}\rangle = \langle g^{-1} \circ \boldsymbol{O}, \boldsymbol{\omega}\rangle + \langle g \circ \boldsymbol{O}, g \circ \boldsymbol{\omega}\rangle,$$

i.e.,

$$\langle g^{-1} \circ \boldsymbol{O}, \boldsymbol{\omega}\rangle = -\langle g \circ \boldsymbol{O}, g \circ \boldsymbol{\omega}\rangle,$$

replace $g^{-1}$ with $g$, then

$$\langle g \circ \boldsymbol{O}, \boldsymbol{\omega}\rangle = -\langle g^{-1} \circ \boldsymbol{O}, g^{-1} \circ \boldsymbol{\omega}\rangle.$$

By definition,

$$k_\lambda(\boldsymbol{x}, \boldsymbol{y}) = \frac{1}{2} \cdot \underset{\boldsymbol{\omega}}{\mathbb{E}}\left[\exp\left((\frac{n-1}{2} - i\lambda)\langle\boldsymbol{x}, \boldsymbol{\omega}\rangle_H + (\frac{n-1}{2} + i\lambda)\langle\boldsymbol{y}, \boldsymbol{\omega}\rangle_H\right)\right]. \tag{5}$$

Now assume, $g \circ \boldsymbol{y} = O$, then consider

$$
\begin{aligned}
k_\lambda(g \circ \boldsymbol{x}, \boldsymbol{O}) &= \frac{1}{2} \cdot \underset{\boldsymbol{\omega}}{\mathbb{E}}\left[\zeta_{\lambda,\boldsymbol{\omega}}(g \circ \boldsymbol{x})^*\zeta_{\lambda,\boldsymbol{\omega}}(\boldsymbol{O})\right] = \frac{1}{2} \cdot \underset{\boldsymbol{\omega}}{\mathbb{E}}\left[\zeta_{\lambda,\boldsymbol{\omega}}(g \circ \boldsymbol{x})^*\right] \\
&= \frac{1}{2} \cdot \underset{\boldsymbol{\omega}}{\mathbb{E}}\left[\exp\left(\left(\frac{n-1}{2} - i\lambda\right)\langle g \circ \boldsymbol{x}, \boldsymbol{\omega}\rangle\right)\right] \\
&= \frac{1}{2} \cdot \underset{\boldsymbol{\omega}}{\mathbb{E}}\left[\exp\left(\left(\frac{n-1}{2} - i\lambda\right)\left(\langle\boldsymbol{x}, g^{-1} \circ \boldsymbol{\omega}\rangle + \langle g \circ \boldsymbol{O}, \boldsymbol{\omega}\rangle\right)\right)\right] \\
&= \frac{1}{2} \cdot \underset{\boldsymbol{\omega}}{\mathbb{E}}\left[\exp\left(\left(\frac{n-1}{2} - i\lambda\right)\left(\langle\boldsymbol{x}, g^{-1} \circ \boldsymbol{\omega}\rangle - \langle g^{-1} \circ \boldsymbol{O}, g^{-1} \circ \boldsymbol{\omega}\rangle\right)\right)\right] \\
&= \frac{1}{2} \cdot \underset{\boldsymbol{\omega}}{\mathbb{E}}\left[\exp\left(\left(\frac{n-1}{2} - i\lambda\right)\left(\langle\boldsymbol{x}, g^{-1} \circ \boldsymbol{\omega}\rangle - \langle\boldsymbol{y}, g^{-1} \circ \boldsymbol{\omega}\rangle\right)\right)\right] \\
&= \frac{1}{2} \cdot \int_{\mathbb{S}^{n-1}} \exp\left(\left(\frac{n-1}{2} - i\lambda\right)\left(\langle\boldsymbol{x}, g^{-1} \circ \boldsymbol{\omega}\rangle - \langle\boldsymbol{y}, g^{-1} \circ \boldsymbol{\omega}\rangle\right)\right) \rho_1(\boldsymbol{\omega})d\boldsymbol{\omega},
\end{aligned}
$$

where $\rho_1(\boldsymbol{\omega})$ is a uniform distribution over the sphere, use $\hat{\boldsymbol{\omega}} = g^{-1} \circ \boldsymbol{\omega}$ as a change of variable, then

$$
\begin{aligned}
k_\lambda(g \circ \boldsymbol{x}, \boldsymbol{O}) &= \int_{\mathbb{S}^{n-1}} \exp\left(\left(\frac{n-1}{2} - i\lambda\right)\left(\langle\boldsymbol{x}, g^{-1} \circ \boldsymbol{\omega}\rangle - \langle\boldsymbol{y}, g^{-1} \circ \boldsymbol{\omega}\rangle\right)\right) \rho_1(\boldsymbol{\omega})d\boldsymbol{\omega} \\
&= \int_{\mathbb{S}^{n-1}} \exp\left(\left(\frac{n-1}{2} - i\lambda\right)\left(\langle\boldsymbol{x}, \hat{\boldsymbol{\omega}}\rangle - \langle\boldsymbol{y}, \hat{\boldsymbol{\omega}}\rangle\right)\right) \rho_1(g \circ \hat{\boldsymbol{\omega}})d(g \circ \hat{\boldsymbol{\omega}}).
\end{aligned}
$$

We claim that the mapping $g$ acts on the boundary with the Jacobian given by

$$\frac{d(g \circ \hat{\boldsymbol{\omega}})}{d(\hat{\boldsymbol{\omega}})} = \frac{1}{2} \cdot \exp((n-1) \cdot \langle g^{-1} \circ \boldsymbol{O}, \hat{\boldsymbol{\omega}}\rangle) = \frac{1}{2} \cdot \left(\frac{1 - \left\|g^{-1} \circ \boldsymbol{O}\right\|^2}{\left\|g^{-1} \circ \boldsymbol{O} - \hat{\boldsymbol{\omega}}\right\|^2}\right)^{n-1}, \tag{6}$$

then

$$k_\lambda(g \circ \boldsymbol{x}, \boldsymbol{O}) = \frac{1}{2} \cdot \int_{\mathbb{S}^{n-1}} \exp\left(\left(\frac{n-1}{2} - i\lambda\right)(\langle \boldsymbol{x}, \hat{\boldsymbol{\omega}}\rangle - \langle \boldsymbol{y}, \hat{\boldsymbol{\omega}}\rangle)\right) \rho_1(g \circ \hat{\boldsymbol{\omega}}) d(g \circ \hat{\boldsymbol{\omega}})$$

$$= \frac{1}{2} \cdot \int_{\mathbb{S}^{n-1}} \exp\left(\left(\frac{n-1}{2} - i\lambda\right)(\langle \boldsymbol{x}, \hat{\boldsymbol{\omega}}\rangle - \langle \boldsymbol{y}, \hat{\boldsymbol{\omega}}\rangle)\right) \rho_1(g \circ \hat{\boldsymbol{\omega}}) \exp((n-1) \cdot \langle g^{-1} \circ \boldsymbol{O}, \hat{\boldsymbol{\omega}}\rangle) d(\hat{\boldsymbol{\omega}})$$

$$= \frac{1}{2} \cdot \int_{\mathbb{S}^{n-1}} \exp\left(\left(\frac{n-1}{2} - i\lambda\right)(\langle \boldsymbol{x}, \hat{\boldsymbol{\omega}}\rangle - \langle \boldsymbol{y}, \hat{\boldsymbol{\omega}}\rangle) + (n-1) \cdot \langle \boldsymbol{y}, \hat{\boldsymbol{\omega}}\rangle\right) \rho_1(g \circ \hat{\boldsymbol{\omega}}) d(\hat{\boldsymbol{\omega}})$$

$$= \frac{1}{2} \cdot \int_{\mathbb{S}^{n-1}} \exp\left(\left(\frac{n-1}{2} - i\lambda\right)\langle \boldsymbol{x}, \hat{\boldsymbol{\omega}}\rangle + \left(n - 1 - \frac{n-1}{2} + i\lambda\right)\langle \boldsymbol{y}, \hat{\boldsymbol{\omega}}\rangle\right) \rho_1(g \circ \hat{\boldsymbol{\omega}}) d(\hat{\boldsymbol{\omega}})$$

$$= \frac{1}{2} \cdot \int_{\mathbb{S}^{n-1}} \exp\left(\left(\frac{n-1}{2} - i\lambda\right)\langle \boldsymbol{x}, \hat{\boldsymbol{\omega}}\rangle + \left(\frac{n-1}{2} + i\lambda\right)\langle \boldsymbol{y}, \hat{\boldsymbol{\omega}}\rangle\right) \rho_1(g \circ \hat{\boldsymbol{\omega}}) d(\hat{\boldsymbol{\omega}}).$$

Since $\rho_1$ is a uniform distribution, this is

$$k_\lambda(g \circ \boldsymbol{x}, \boldsymbol{O}) = \frac{1}{2} \cdot \mathbb{E}_{\hat{\boldsymbol{\omega}}}\left[\exp\left(\left(\frac{n-1}{2} - i\lambda\right)\langle \boldsymbol{x}, \hat{\boldsymbol{\omega}}\rangle + \left(\frac{n-1}{2} + i\lambda\right)\langle \boldsymbol{y}, \hat{\boldsymbol{\omega}}\rangle\right)\right],$$

compared with Equation 5, it follows that $k_\lambda(g \circ \boldsymbol{x}, \boldsymbol{O}) = k_\lambda(\boldsymbol{x}, \boldsymbol{y})$. Since $k_\lambda(g \circ \boldsymbol{x}, \boldsymbol{O})$ only depends on $d_{\mathbb{H}}(g \circ \boldsymbol{x}, \boldsymbol{O}) = d_{\mathbb{H}}(\boldsymbol{x}, g^{-1} \circ \boldsymbol{O}) = d_{\mathbb{H}}(\boldsymbol{x}, \boldsymbol{y})$ from Lemma A.3, then $k_\lambda(\boldsymbol{x}, \boldsymbol{y})$ is distance-invariant, and hence isometry-invariant.

It suffices to prove Equation 6, i.e.,

$$\frac{d(g \circ \boldsymbol{\omega})}{d(\boldsymbol{\omega})} = \left(\frac{1 - \left\|g^{-1} \circ \boldsymbol{O}\right\|^2}{\left\|g^{-1} \circ \boldsymbol{O} - \boldsymbol{\omega}\right\|^2}\right)^{n-1},$$

clearly it holds when $g$ is an rotation, it suffices to show this for a translation isometry. Denote $\mathrm{Inv}(\boldsymbol{x}) = \frac{\boldsymbol{x}}{\|\boldsymbol{x}\|^2}$, then in the Poincarè Ball model, all translation isometry takes the form

$$T_{\boldsymbol{a}}(\boldsymbol{x}) = -\boldsymbol{a} + (1 - \|\boldsymbol{a}\|^2)\,\mathrm{Inv}(\mathrm{Inv}(\boldsymbol{x}) - \boldsymbol{a}),$$

where both $\boldsymbol{x}, \boldsymbol{a}$ in the Poincarè Ball model and $T_{\boldsymbol{a}}(\boldsymbol{a}) = \boldsymbol{O}, T_{\boldsymbol{a}}^{-1}(\boldsymbol{O}) = \boldsymbol{a}$. Thus,

$$T_{\boldsymbol{a}}(\boldsymbol{\omega}) = -\boldsymbol{a} + (1 - \|\boldsymbol{a}\|^2)\,\mathrm{Inv}(\mathrm{Inv}(\boldsymbol{\omega}) - \boldsymbol{a}) = -\boldsymbol{a} + (1 - \|\boldsymbol{a}\|^2)\,\mathrm{Inv}(\boldsymbol{\omega} - \boldsymbol{a}) = -\boldsymbol{a} + (1 - \|\boldsymbol{a}\|^2)\frac{\boldsymbol{\omega} - \boldsymbol{a}}{\|\boldsymbol{\omega} - \boldsymbol{a}\|^2},$$

where we use the fact $\|\boldsymbol{\omega}\| = 1$. Since the integral is taken over the unit sphere with $\|\boldsymbol{\omega}\| = 1, \|T_{\boldsymbol{a}}(\boldsymbol{\omega})\| = 1$, we consider only the mapping of $T_{\boldsymbol{a}}$ restricted to the first $n - 1$ (free) dimensions, with an abuse of notation, regard $\boldsymbol{\omega}$ as an $n - 1$ dimensional vector. Then the Jacobian of $T_{\boldsymbol{a}}(\boldsymbol{\omega})$ with respect to $\boldsymbol{\omega}$ is

$$dT_{\boldsymbol{a}}(\boldsymbol{\omega}) = (1 - \|\boldsymbol{a}\|^2)d\left(\frac{\boldsymbol{\omega} - \boldsymbol{a}}{\|\boldsymbol{\omega} - \boldsymbol{a}\|^2}\right),$$

we can calculate that

$$d\left(\frac{\boldsymbol{\omega} - \boldsymbol{a}}{\|\boldsymbol{\omega} - \boldsymbol{a}\|^2}\right) = \frac{\|\boldsymbol{\omega} - \boldsymbol{a}\|^2 d\boldsymbol{\omega} - (\boldsymbol{\omega} - \boldsymbol{a}) \cdot 2(\boldsymbol{\omega} - \boldsymbol{a})^\mathsf{T} d\boldsymbol{\omega}}{\|\boldsymbol{\omega} - \boldsymbol{a}\|^4}$$

$$= \frac{1}{\|\boldsymbol{\omega} - \boldsymbol{a}\|^2} \cdot \frac{\|\boldsymbol{\omega} - \boldsymbol{a}\|^2 - 2(\boldsymbol{\omega} - \boldsymbol{a}) \cdot (\boldsymbol{\omega} - \boldsymbol{a})^\mathsf{T}}{\|\boldsymbol{\omega} - \boldsymbol{a}\|^2} d\boldsymbol{\omega}$$

$$= \frac{d\boldsymbol{\omega}}{\|\boldsymbol{\omega} - \boldsymbol{a}\|^2} \cdot \left(\mathbf{I}_{n-1,n-1} - \frac{2(\boldsymbol{\omega} - \boldsymbol{a}) \cdot (\boldsymbol{\omega} - \boldsymbol{a})^\mathsf{T}}{\|\boldsymbol{\omega} - \boldsymbol{a}\|^2}\right),$$

then

$$\frac{dT_{\boldsymbol{a}}(\boldsymbol{\omega})}{d\boldsymbol{\omega}} = \frac{1 - \|\boldsymbol{a}\|^2}{\|\boldsymbol{\omega} - \boldsymbol{a}\|^2}\left(\mathbf{I}_{n-1,n-1} - \frac{2(\boldsymbol{\omega} - \boldsymbol{a}) \cdot (\boldsymbol{\omega} - \boldsymbol{a})^\mathsf{T}}{\|\boldsymbol{\omega} - \boldsymbol{a}\|^2}\right),$$

note the relation that

$$\det(\mathbf{I} + \boldsymbol{x}\boldsymbol{y}^\mathsf{T}) = 1 + \boldsymbol{x}^\mathsf{T}\boldsymbol{y},$$

then

$$\det(\mathbf{I}_{n-1,n-1} - \frac{2(\boldsymbol{\omega} - \boldsymbol{a}) \cdot (\boldsymbol{\omega} - \boldsymbol{a})^{\mathsf{T}}}{\|\boldsymbol{\omega} - \boldsymbol{a}\|^2}) = 1 - \frac{2(\boldsymbol{\omega} - \boldsymbol{a})^{\mathsf{T}} \cdot (\boldsymbol{\omega} - \boldsymbol{a})}{\|\boldsymbol{\omega} - \boldsymbol{a}\|^2} = 1 - 2 = -1,$$

then the absolute value of determinant of the Jacobian is

$$|\det(\frac{dT_{\boldsymbol{a}}(\boldsymbol{\omega})}{d\boldsymbol{\omega}})| = |\det\left(\frac{1 - \|\boldsymbol{a}\|^2}{\|\boldsymbol{\omega} - \boldsymbol{a}\|^2}(\mathbf{I}_{n-1,n-1} - \frac{2(\boldsymbol{\omega} - \boldsymbol{a}) \cdot (\boldsymbol{\omega} - \boldsymbol{a})^{\mathsf{T}}}{\|\boldsymbol{\omega} - \boldsymbol{a}\|^2})\right)|$$

$$= \left(\frac{1 - \|\boldsymbol{a}\|^2}{\|\boldsymbol{\omega} - \boldsymbol{a}\|^2}\right)^{n-1}$$

$$= \left(\frac{1 - \|g^{-1} \circ \boldsymbol{O}\|^2}{\|\boldsymbol{\omega} - g^{-1} \circ \boldsymbol{O}\|^2}\right)^{n-1},$$

with which a change of variable would give

$$\frac{d(g \circ \boldsymbol{\omega})}{d(\boldsymbol{\omega})} = \left(\frac{1 - \|g^{-1} \circ \boldsymbol{O}\|^2}{\|g^{-1} \circ \boldsymbol{O} - \boldsymbol{\omega}\|^2}\right)^{n-1},$$

which finishes the proof. $\square$

***Proof of Theorem 4.1.*** As a result of Lemma A.3 and Lemma A.4, the expression for $k_\lambda(\boldsymbol{x}, \boldsymbol{y})$ follows:

$$k_\lambda(\boldsymbol{x}, \boldsymbol{y}) = \frac{1}{2} \cdot {}_2F_1\left(\frac{n-1}{2} + i\lambda, \frac{n-1}{2} - i\lambda; \frac{n}{2}; \frac{1}{2}(1 - \cosh(d_{\mathbb{H}}(\boldsymbol{x}, \boldsymbol{y})))\right),$$

where ${}_2F_1$ is the hypergeometric function, we can also apply the Euler transformation to get

$$k_\lambda(\boldsymbol{x}, \boldsymbol{y}) = \frac{1}{2} \cdot \left(\frac{1}{2}(1 + \cosh(d_{\mathbb{H}}(\boldsymbol{x}, \boldsymbol{y})))\right)^{1 - \frac{n}{2}} \cdot {}_2F_1\left(\frac{1}{2} + i\lambda, \frac{1}{2} - i\lambda; \frac{n}{2}; \frac{1}{2}(1 - \cosh(d_{\mathbb{H}}(\boldsymbol{x}, \boldsymbol{y})))\right).$$

If we let

$$z = \frac{1}{2}(1 - \cosh(d_{\mathbb{H}}(\boldsymbol{x}, \boldsymbol{y})))$$

then this is written more succinctly as

$$k_\lambda(\boldsymbol{x}, \boldsymbol{y}) = \frac{1}{2} \cdot (1 - z)^{1 - \frac{n}{2}} \cdot {}_2F_1\left(\frac{1}{2} + i\lambda, \frac{1}{2} - i\lambda; \frac{n}{2}; z\right).$$

We can also write this as a Legendre function,

$$k_\lambda(\boldsymbol{x}, \boldsymbol{y}) = \frac{1}{2} \cdot (1 - z)^{1 - \frac{n}{2}} \cdot \Gamma\left(\frac{n}{2}\right) \cdot z^{\frac{2-n}{4}} \cdot (1 - z)^{\frac{n-2}{4}} \cdot P_{-\frac{1}{2} + i\lambda}^{1 - \frac{n}{2}}(1 - 2z)$$

$$= \frac{1}{2} \cdot (z(1 - z))^{\frac{2-n}{4}} \cdot \Gamma\left(\frac{n}{2}\right) \cdot P_{-\frac{1}{2} + i\lambda}^{1 - \frac{n}{2}}(1 - 2z)$$

Observe that

$$z(1 - z) = \frac{1}{4}(1 - \cosh^2(d_{\mathbb{H}}(\boldsymbol{x}, \boldsymbol{y}))) = \frac{1}{4}(-\sinh^2(d_{\mathbb{H}}(\boldsymbol{x}, \boldsymbol{y})))$$

and similarly

$$1 - 2z = \cosh(d_{\mathbb{H}}(\boldsymbol{x}, \boldsymbol{y})).$$

This is manifestly real because

$$k_\lambda(\boldsymbol{x}, \boldsymbol{y}) = \frac{1}{2} \cdot \sum_{k=0}^{\infty} \prod_{m=0}^{k-1} \frac{\left(\frac{n-1}{2} + m + i\lambda\right)\left(\frac{n-1}{2} + m - i\lambda\right)}{\left(\frac{n}{2} + m\right)(1 + m)} \cdot \frac{1}{2}(1 - \cosh(d_{\mathbb{H}}(\boldsymbol{x}, \boldsymbol{y})))$$

$$= \frac{1}{2} \cdot \sum_{k=0}^{\infty} \prod_{m=0}^{k-1} \frac{\left(\frac{n-1}{2} + m\right)^2 + \lambda^2}{\left(\frac{n}{2} + m\right)(1 + m)} \cdot \frac{1}{2}(1 - \cosh(d_{\mathbb{H}}(\boldsymbol{x}, \boldsymbol{y}))).$$

If we draw HyLa features using a distribution $\rho$ over $\lambda$, then the resulting approximated kernel will be

$$
\begin{aligned}
k(\boldsymbol{x}, \boldsymbol{y}) &= \frac{1}{2} \cdot \int_{-\infty}^{\infty} k_\lambda(\boldsymbol{x}, \boldsymbol{y}) \cdot \rho(\lambda) \, d\lambda \\
&= \frac{1}{2} \cdot \int_{-\infty}^{\infty} {}_2F_1\left(\frac{n-1}{2} + i\lambda, \frac{n-1}{2} - i\lambda; \frac{n}{2}; \frac{1}{2}\left(1 - \cosh(d_{\mathbb{H}}(\boldsymbol{x}, \boldsymbol{y}))\right)\right) \cdot \rho(\lambda) \, d\lambda.
\end{aligned}
$$

$\square$

*Proof of Theorem 4.2*. Denote

$$
\phi_\lambda(\boldsymbol{z}) = \int_{\mathbb{S}^{n-1}} \zeta_{\lambda, \boldsymbol{\omega}}(\boldsymbol{z}) d\boldsymbol{\omega} = \int_{\mathbb{S}^{n-1}} \exp\left((\frac{n-1}{2} + i\lambda)\langle \boldsymbol{\omega}, \boldsymbol{z} \rangle_H\right) d\boldsymbol{\omega},
$$

which are basic spherical functions. For any $\boldsymbol{x}, \boldsymbol{y} \in \mathcal{B}^n$, assume $g_{\boldsymbol{y}}$ is an isometry that maps $\boldsymbol{y}$ to the origin, i.e., $g_{\boldsymbol{y}} \circ \boldsymbol{y} = O$, denote $\hat{\boldsymbol{x}} = g_{\boldsymbol{y}} \circ \boldsymbol{x}$, then $k_\lambda(\boldsymbol{x}, \boldsymbol{y}) = k_\lambda(\hat{\boldsymbol{x}}, O)$ for any $\lambda$, note that

$$
\begin{aligned}
k_\lambda(\hat{\boldsymbol{x}}, O) &= \frac{1}{2} \cdot \int_{\mathbb{S}^{n-1}} \exp\left((\frac{n-1}{2} + i\lambda)\langle \boldsymbol{\omega}, \hat{\boldsymbol{x}} \rangle_H\right) \rho_1(\boldsymbol{\omega}) d\boldsymbol{\omega} \\
&= \frac{1}{2} \cdot \frac{1}{\text{Area}(\mathbb{S}^{n-1})} \int_{\mathbb{S}^{n-1}} \exp\left((\frac{n-1}{2} + i\lambda)\langle \boldsymbol{\omega}, \hat{\boldsymbol{x}} \rangle_H\right) d\boldsymbol{\omega} \\
&= \frac{1}{2} \cdot \frac{1}{\text{Area}(\mathbb{S}^{n-1})} \phi_\lambda(\hat{\boldsymbol{x}}),
\end{aligned}
$$

where we use the fact that $\rho_1(\boldsymbol{\omega})$ is a uniform distribution over the sphere. Assume the existence of an associated density $\rho(\lambda)$ with the kernel, then

$$
\begin{aligned}
k(\boldsymbol{x}, \boldsymbol{y}) = k(d_{\mathbb{H}}(\boldsymbol{x}, \boldsymbol{y})) &= k(d_{\mathbb{H}}(\hat{\boldsymbol{x}}, O)) \\
&= \int_{-\infty}^{\infty} k_\lambda(\boldsymbol{x}, \boldsymbol{y}) \cdot \rho(\lambda) \, d\lambda \\
&= \int_{-\infty}^{\infty} k_\lambda(\hat{\boldsymbol{x}}, O) \cdot \rho(\lambda) \, d\lambda \\
&= \frac{1}{2} \cdot \int_{-\infty}^{\infty} \frac{1}{\text{Area}(\mathbb{S}^{n-1})} \phi_\lambda(\hat{\boldsymbol{x}}) \cdot \rho(\lambda) \, d\lambda \\
&= \frac{1}{2} \cdot \frac{1}{\text{Area}(\mathbb{S}^{n-1})} \int_{-\infty}^{\infty} \phi_\lambda(\hat{\boldsymbol{x}}) \cdot \rho(\lambda) \, d\lambda \\
&= \frac{1}{\pi \text{Area}(\mathbb{S}^{n-1})} \int_{-\infty}^{\infty} \frac{\rho(\lambda)}{\lambda \tanh\left(\frac{\pi\lambda}{2}\right)} \cdot \phi_\lambda(\hat{\boldsymbol{x}}) \cdot |c(\lambda)|^{-2} \, d\lambda
\end{aligned}
$$

where $|c(\lambda)|^{-2} = \frac{\pi\lambda}{2} \tanh \frac{\pi\lambda}{2}$ when $\lambda \in \mathbb{R}$. Note that the last integral is exactly the inverse spherical transform (Helgason, 2022) of $\frac{\rho(\lambda)}{\lambda \tanh\left(\frac{\pi\lambda}{2}\right)}$, hence it can be derived in the reverse direction by taking the spherical transform of $k(d_{\mathbb{H}}(\hat{\boldsymbol{x}}, O))$, i.e.,

$$
\frac{\rho(\lambda)}{\lambda \tanh\left(\frac{\pi\lambda}{2}\right)} = \int_{\mathcal{B}^n} k(d_{\mathbb{H}}(\hat{\boldsymbol{x}}, O)) \phi_{-\lambda}(\hat{\boldsymbol{x}}) d\hat{\boldsymbol{x}}.
$$

Hence $\rho(\lambda)$ can be derived as

$$
\begin{aligned}
\rho(\lambda) &= \lambda \tanh\left(\frac{\pi\lambda}{2}\right) \int_{\mathcal{B}^n} k(d_{\mathbb{H}}(\hat{\boldsymbol{x}}, O)) \phi_{-\lambda}(\hat{\boldsymbol{x}}) d\hat{\boldsymbol{x}} \\
&= \lambda \tanh\left(\frac{\pi\lambda}{2}\right) \int_{\mathcal{B}^n} \int_{\partial\mathcal{B}^n} k(d_{\mathbb{H}}(\boldsymbol{z}, O)) \exp\left((\frac{n-1}{2} - i\lambda)\langle \boldsymbol{\omega}, \boldsymbol{z} \rangle_H\right) d\boldsymbol{\omega} d\boldsymbol{z}.
\end{aligned}
$$

Therefore, given an isometry-invariant positive semidefinite kernel $k(\boldsymbol{x}, \boldsymbol{y}) = k(d_{\mathbb{H}}(\boldsymbol{x}, \boldsymbol{y}))$, we can compute $\rho(\lambda)$ following the above expression if it exists, then the rest of the theorem follows. $\square$

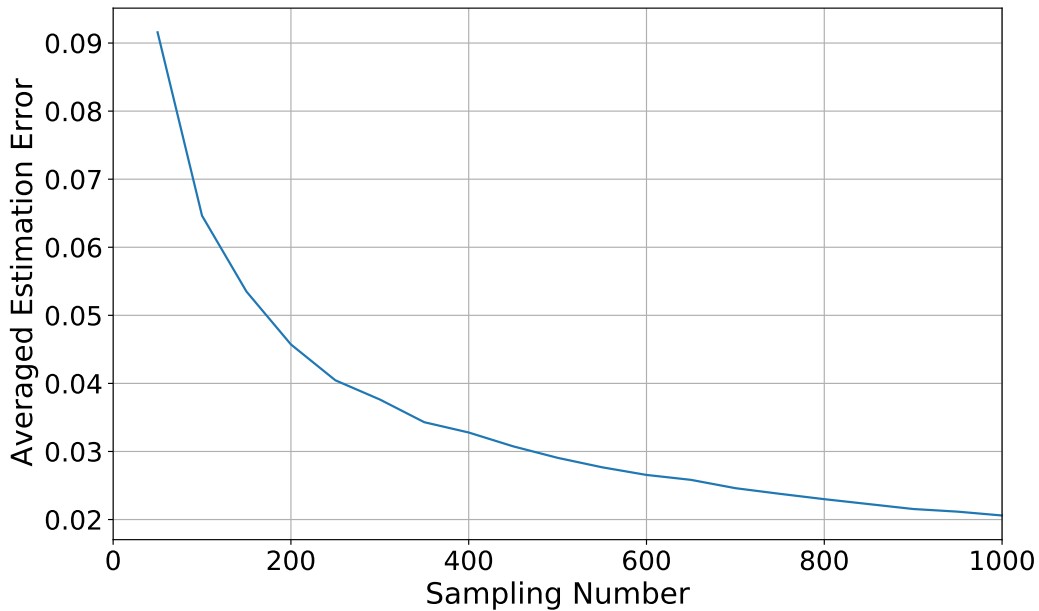

Figure 6: Averaged estimation error of HyLa to the kernel

## B  CONCENTRATION OF THE KERNEL ESTIMATION

The readers may wonder whether there is a concentration behavior using the random variable $\langle \phi(\boldsymbol{x}), \phi(\boldsymbol{y}) \rangle$ to approximate $k(\boldsymbol{x}, \boldsymbol{y})$. Unfortunately, the eigenfunction $\phi(\boldsymbol{x})$ itself is not a sub-Gaussian so as to derive a concentration bound in a straightforward way, but we do provide a numerical experiment to measure the estimation behavior. For the estimation in Figure 3, we sampled $1,000$ different eigenfunctions.

We first fix a set of $1,000$ points $x_i$ by uniformly sampling over 2-dimensional hyperbolic space, then approximate the kernel $k(\boldsymbol{o}, \boldsymbol{x_i})$ using $\langle \phi(\boldsymbol{x_i}), \phi(\boldsymbol{o}) \rangle$ by sampling with an increasing number of eigenfunctions, ranging from $50$ to $1,000$. At last, we compute the mean absolute error of the estimation $\langle \phi(\boldsymbol{x_i}), \phi(\boldsymbol{o}) \rangle$ to the true kernel value $k(\boldsymbol{o}, \boldsymbol{x_i})$. We plot the mean estimation error in Figure 6, which seems to be an exponentially decay, it's an interesting future work to investigate this estimation error.

## C  NODE EMBEDDING VS FEATURE EMBEDDING

When HyLa is adopted at node level, i.e., each vertex/node $v_i$ in the graph is associated with a hyperbolic embedding parameter $\boldsymbol{z}_i \in \mathcal{B}^{d_0}$. Then the inner product of HyLa features $\langle \phi(\boldsymbol{z}_i), \phi(\boldsymbol{z}_j) \rangle$ of vertex $v_i, v_j$ approximates some kernel $k(\boldsymbol{z}_i, \boldsymbol{z}_j)$. The optimization of $z_i$ encourages learning of the kernel on the hyperbolic space to solve the task.

When HyLa is adopted at feature level, i.e., each column dimension of the node feature $\mathbf{X} \in \mathbb{R}^{n \times d}$ is associated with a hyperbolic embedding parameter $\boldsymbol{z}_i \in \mathcal{B}^{d_0}$. The HyLa feature associated to each vertex/node $v_i$ is then computed as $\sum_{k=1}^{d} \mathbf{X}_{ik} \phi(\boldsymbol{z}_k)$, where $\sum_{k=1}^{d} \mathbf{X}_{ik} = 1$ if a row-normalization is applied on the original node features.

Therefore, the inner product of two node HyLa features is

$$\langle \sum_{k=1}^{d} \mathbf{X}_{ik} \phi(\boldsymbol{z}_k), \sum_{l=1}^{d} \mathbf{X}_{jl} \phi(\boldsymbol{z}_l) \rangle = \sum_{k,l=1}^{d} \mathbf{X}_{ik} \mathbf{X}_{jl} \langle \phi(\boldsymbol{z}_k), \phi(\boldsymbol{z}_l) \rangle,$$

Table 3: Node classification Dataset statistics.

| Setting | Dataset | # Nodes | # Edges | Classes | Features |
|---------|---------|---------|---------|---------|----------|
| Trans-ductive | Cora | 2,708 | 5,429 | 7 | 1,433 |
| | Citeseer | 3,327 | 4,732 | 6 | 3,703 |
| | Pubmed | 19,717 | 44,338 | 3 | 500 |
| | Disease | 1,044 | 1,043 | 2 | 1,000 |
| | Airport | 3,188 | 18,631 | 4 | 4 |
| Inductive | Reddit | 233K | 11.6M | 41 | 602 |

Table 4: Text classification Dataset statistics.

| Dataset | # Docs | # Words | Average Length | Classes |
|---------|--------|---------|----------------|---------|
| R8 | 7,674 | 7688 | 65.72 | 8 |
| R52 | 9,100 | 8892 | 69.82 | 52 |
| Ohsumed | 7400 | 14157 | 135.82 | 23 |
| MR | 10662 | 18764 | 20.39 | 2 |

which in expectation equals a linear combination of kernels, i.e., $\sum_{k,l=1}^{d} \mathbf{X}_{ik}\mathbf{X}_{jl}k(\boldsymbol{z}_k, \boldsymbol{z}_l)$. Therefore, it captures a much more complicated kernel relation on the hyperbolic space than directly embedding nodes.

## D  TWO-STEP APPROACH

For the purpose of end-to-end learning, in our experiments, we jointly learn the embedding parameter $\mathbf{Z}$ and weight $\mathbf{W}$ in SGC during the training time, as detailed in subsection E.2. It's possible to adopt a two-step approach, i.e., first pretrain a hyperbolic embedding, then fix the embedding and train the graph learning model only. In the first step, for example, optimization-based methods (Nickel & Kiela, 2017; 2018) and combinatorial construction methods (Sala et al., 2018; Sonthalia & Gilbert, 2020) can be adopted by supervising the graph connectivity. However, these methods only utilize the graph structure information, but ignore the node feature information $\mathbf{X}$, which leads to a natural performance degradation. In comparison, as shown in experiments and analyzed in Appendix C, our end-to-end learning of HyLa can be used to embed features and enables learning a complex kernel representation to avoid this shortcoming. Intuitively, the graph connectivity information can be too general for downstreaming tasks which rely more on semantic information. It's not clear to us how to encode the semantic information (node features) into embedding following e.g., (Nickel & Kiela, 2017; 2018).

Another way (Chami et al., 2019) of deriving a pretrained hyperbolic embedding that might take semantic information into consideration is to train a link prediction model, however, this method is not efficient as HGCN, shown in Figure 5.

## E  EXPERIMENT DETAILS

### E.1  TASK AND DATASET

We provide a detailed description/table of used datasets in Table 3 and Table 4.

1. **Citation Networks.** Cora, Citeseer and Pubmed Sen et al. (2008) are standard citation network benchmarks, where nodes represent papers, connected to each other via citations. We follow the standard splits Kipf & Welling (2016) with 20 nodes per class for training, 500 nodes for validation and 1000 nodes for test.
2. **Disease propagation tree** Chami et al. (2019). This is tree networks simulating the SIR disease spreading model Anderson & May (1992), where the label is whether a node was infected or not and the node features indicate the susceptibility to the disease. We use dataset splits of $30/10/60\%$ for train/val/test set.
3. **Airport.** We take this dataset from Chami et al. (2019). This is a transductive dataset where nodes represent airports and edges represent the airline routes as from OpenFlights. Airport contains 3,188 nodes, each node has a 4 dimensional feature representing geographic information (longitude, latitude and altitude), and GDP of the country where the airport belongs to. For node classification, labels are chosen to be the population of the country where the airport belongs to. We use dataset splits of $524/524$ nodes for val/test set.
4. **Reddit.** This is a much larger graph dataset built from Reddit posts, where the label is the community, or "subreddit", that a post belongs to. Two nodes are connected if the same user comments on both. We use a dataset split of $152K/24K/55K$ follows Hamilton et al. (2017); Chen et al. (2018), similarly, we evaluate HyLa inductively by following Wu et al.

Table 5: Hyper-parameters for node classification.

| Dataset | $d_0$ | $d_1$ | $K$ | $s$ | $lr_1$ | $lr_2$ | # Epochs |
|---------|-------|-------|-----|-----|--------|--------|----------|
| Disease | 16 | 100 | 5 | 1.0 | 0.05 | 0.0001 | 100 |
| Airport | 16 | 1000 | 2 | 0.01 | 0.5 | 0.1 | 100 |
| Pubmed | 16 | 100 | 5 | 0.1 | 0.5 | 0.001 | 200 |
| Citeseer | 16 | 500 | 5 | 1.0 | 0.1 | 0.001 | 100 |
| Cora | 16 | 100 | 2 | 1.0 | 0.1 | 0.01 | 100 |
| Reddit | 50 | 1000 | 2 | 0.5 | 0.1 | 0.001 | 100 |

Table 6: Hyper-parameters for text classification.

| Dataset | Transductive Setting | | | | | Inductive Setting | | | | |
|---------|-------|-------|-----|--------|--------|-------|-------|-----|--------|--------|
| | $d_0$ | $d_1$ | $s$ | $lr_1$ | $lr_2$ | $d_0$ | $d_1$ | $s$ | $lr_1$ | $lr_2$ |
| R8 | 50 | 500 | 0.5 | 0.01 | 0.0001 | 50 | 500 | 0.5 | 0.001 | 0.0001 |
| R52 | 50 | 500 | 0.5 | 0.1 | 0.0001 | 50 | 1000 | 0.5 | 0.008 | 0.0001 |
| Ohsumed | 50 | 500 | 0.5 | 0.01 | 0.0001 | 50 | 1000 | 0.1 | 0.001 | 0.0001 |
| MR | 30 | 500 | 0.5 | 0.1 | 0.0001 | 50 | 500 | 0.5 | 0.01 | 0.0001 |

(2019): we train on a subgraph comprising only training nodes and test with the original graph.

### E.2 TRAINING DETAILS.

We use HyLa together with SGC model as softmax($\mathbf{A}^K \overline{\mathbf{X}} \mathbf{W}$), where the HyLa feature matrix $\overline{\mathbf{X}} \in \mathbb{R}^{n \times d_1}$ is derived from the hyperbolic embedding $\mathbf{Z} \in \mathbb{R}^{n \times d_0}$ using Algorithm 1. Specifically, we randomly sample constants of HyLa features $\overline{\mathbf{X}}$ by sampling the boundary points $\boldsymbol{\omega}$ uniformly from the boundary $\partial \mathcal{B}^n$, eigenvalue constants $\lambda$ from a zero-mean $s$-standard-deviation Gaussian and biases $b$ uniformly from $[0, 2\pi]$. These constants remain fixed throughout training.

We use cross-entropy as the loss function and jointly optimize the low dimensional hyperbolic embedding $\mathbf{Z}$ and linear weight $\mathbf{W}$ simultaneously during training. Specifically, Riemannian SGD optimizer Bonnabel (2013) (of learning rate $lr_1$) for $\mathbf{Z}$ and Adam Kingma & Ba (2014) optimizer (of learning rate $lr_2$) for $\mathbf{W}$. RSGD naturally scales to very large graph because the graph connectivity pattern is sufficiently sparse. We adopt early-stopping as regularization. We tune the hyper-parameter via grid search over the parameter space. Each hyperbolic embedding is initialized around the origin, by sampling each coordinate at random from $[-10^{-5}, 10^{-5}]$.

### E.3 HYPER-PARAMETERS

We provide the detailed values of hyper-parameters for **node classification** and **text classification** in Table 5 and Table 6 respectively. Particularly, we fix $K = 2$ for the text classification task and train the model for a maximum of 200 epochs without using any regularization (e.g. early stopping). Also note that in the transductive text classification setting, HyLa is used at node level, hence the size of parameters will be proportional to the size of graph, in which case, $d_0$ and $d_1$ can not be too large so as to avoid OOM. In the inductive text classification setting, there is no such constraint as the dimension of lower level features is not very large itself. Please check the code for more details.

## F TIMING

We show the specific training timing statistics of different models on Pubmed dataset in Table 7. Particularly for HGCN model, in order to achieve the report performances, we follow the same training procedure using public code, which is divided into two stages: (1) a link prediction task on the dataset to learn hyperbolic embeddings, and (2) use the pretrained embeddings to train a MLP classifier. Hence, we add the timing of both stages as the timing for HGCN.

Table 7: Training time on Pubmed.

| Model | Timing (seconds) |
|---|---|
| SGC | 0.37 |
| GCN | 1.25 |
| GAT | 12.52 |
| HNN | 2.41 |
| HGCN | 15.41 |
| LGCN | 10.93 |
| HyLa-SGC | 3.51 |

