# OpenReview forum: "Random Laplacian Features for Learning with Hyperbolic Space"
_ICLR.cc/2023/Conference — ICLR 2023 poster_

### Official Review · Reviewer_BVuW · 2022-10-23

**Confidence:** 3
**Correctness:** 3
**Technical Novelty And Significance:** 3
**Empirical Novelty And Significance:** 3
**Recommendation:** 6

**Clarity, Quality, Novelty And Reproducibility:**

I think the method description and the introduction to the main theoretical concepts can be better introduced, particularly for non-expert readers. I appreciate the quality of the work and the novelty, while I think some effort might be required to reproduce the results of the paper. I cannot find any statement about code release.

**Strength And Weaknesses:**

STRENGTH
=======
1) SIMPLICITY: While I do not fully understand some parts of the method, the overall principles seem straightforward. I expect that with some better details about the implementation, the method can be easily reproducible. Adopting Laplacian features as input of a network is a well-known approach to inject non-euclidean knowledge inside the network, but I am not aware of other works that compute such features in hyperbolic spaces.

2) EXPERIMENTS: The method is tested on different kinds of graphs from different families and structures. The results suggest that in the worst case, the method can be applied without harm, while in many cases, it improves.

WEAKNESSES
=======
1) PRESENTATION: as a non-expert in hyperbolic spaces, I had a hard time understanding exactly some parts of the method. For example, it is not clear to me how "a low dimensional hyperbolic embedding is initialized for each node". I think a plain explaination might help the reader and increase the usability of the paper methodology

2) APPLICABILITY AND LIMITATIONS: It is not completely clear to me if the method can have an impact in different applications from node classification since all the experiments are performed in this context. Exploiting isometric invariance has been widely popular in contexts like 3D meshes to compute correspondence between two objects. I wonder if the same holds here. Also, while the method indeed improves from euclidean networks, it comes with non-trivial computational costs (i.e., from Figure 5, SGC is 10x times faster). Finally, in Conclusions Section, it is mentioned that in future, a direction is to adopt "more numerically stable representations of the hyperbolic embeddings to avoid potential 'NaN problems'". I would like to know more about these issues and where they arise.

**Summary Of The Paper:**

The paper outline a method to use hyperbolic spaces in a graph neural network. The approach mainly relies on the computation of Laplacian Eigenfunctions: after embedding the graph nodes in some space, they are mapped via a kernel transformation to their hyperbolic features (i.e., Hyperbolic Laplacian eigenfunctions), and lastly, a standard euclidean network can be used. The method shows comparable results with the SoTA and a performance gain over previous hyperbolic networks.

**Summary Of The Review:**

I find the proposed methodology interesting and in line with recent research directions. As a non-expert, I would prefer better insights and explanations to fully appreciate the impact of this work, but I think the highlighted weaknesses can be addressed before the submission. Looking forward to the rebuttal to clarify my doubts.

---

> ### Author Response · Authors · 2022-11-18
> **Response to Reviewer BVuW**
>
> We thank the reviewer for considering our work as novel, simple, straightforward and recognize our experiment efforts, please see below for answers to your questions and suggestions:
>
> Q1: how "a low dimensional hyperbolic embedding is initialized for each node"?
>
> We updated sec. 5 of the paper to make it clear and included discussion on this in Appendix D. For the purpose of end-to-end learning, in our experiments, we jointly learn the embedding parameter $Z$ and weight $W$ in SGC during the training time (The optimization of $Z$ is actually where the 10x times computation cost than SGC mainly comes from). It’s also possible to use a pretrained hyperbolic embedding from some tasks then fix it during training time. We provide a discussion on this in Appendix D.
>
> Q2: APPLICABILITY AND LIMITATIONS:
>
> Our proposed random feature is defined for general hyperbolic space, it’s certainly an interesting direction to study the usage of HyLa on other tasks. We are not familiar with the 3D meshes area, but to our knowledge, it’s a headache for hyperbolic models to extract meaningful Euclidean features from hyperbolic embeddings for further processing; most existing models adopt the $\log$ map as analyzed in sec. 2, while HyLa provides such a meaningful Euclidean feature.
>
> It is possible to use the HyLa function as the mapping (e.g., in place of $\log$) in many hyperbolic models/frameworks, for example, 3D skeleton-based action recognition [1] and deep reinforcement learning [2]. There is work exploiting the isometric invariance in hyperbolic space on some other graph tasks, such as knowledge graphs where the relations are modeled with hyperbolic isometries.
>
> The computation cost arises from optimizing hyperbolic embedding parameters for the goal of end-to-end training. However, as mentioned, with a pretrained embedding given, the computation cost would be nearly the same as SGC. The NaN problem [3] happens due to the imprecision issue when representing hyperbolic space with floating numbers in practice: practitioners can easily run into NaNs during training a hyperbolic model. A small floating point representation error can cause a large distortion to the representation of the space particularly at points far away from the origin in hyperbolic space. Some analysis and robust solutions such as [1] were proposed to work around this problem.
>
> Question and suggestions in Clarity:
>
> Due to page limits, we can only add/modify part of introduction/discussion on the mathematical aspects. Just for the reviewer’s reference, our code is provided in the supplementary material at submission time.
>
> [1] Peng, Wei and Shi, Jingang and Xia, Zhaoqiang and Zhao, Guoying, Mix dimension in poincar{\'e} geometry for 3d skeleton-based action recognition, ACM MM 2020
> [2] Edoardo Cetin, Benjamin Chamberlain, Michael Bronstein, Jonathan J Hunt, Hyperbolic deep reinforcement learning, 2022.
> [3] Tao Yu and Chris De Sa. Numerically accurate hyperbolic embeddings using tiling-based models. NeurIPS 2019.

---

> > ### Comment · Reviewer_BVuW · 2022-11-23
> > **Post-Rebuttal**
> >
> > I thank the authors for their response.
> >
> > Initially, my concerns were about the presentation, and some details could have been better presented in the text (as also noticed by Reviewer Ue38). The authors address such a point quite well, and I consider it solved.
> >
> > I see a disagreement from other reviews: Reviewer iAHJ criticizes the experiments with a quite harsh score. However, the review is particularly poor in terms of arguments, does not provide precise details about the observed "strange results", and fails to offer concrete suggestions. If Reviewer iAHJ doesn't discuss further, I think the Author's response address the point quite well, and I am satisfied.
> >
> > For these reasons, I would keep my score, leaning toward acceptance. At the same time, for the final version, I highly suggest including all the needed clarifications (also adding other appendix sections if needed) and easing the accessibility of the paper with a softer introduction to the field.

---

### Official Review · Reviewer_Ue38 · 2022-10-25

**Confidence:** 3
**Correctness:** 3
**Technical Novelty And Significance:** 4
**Empirical Novelty And Significance:** 2
**Recommendation:** 8

**Clarity, Quality, Novelty And Reproducibility:**

**Clarity**
---

Besides the above mentioned weakness about the missing discussion on the how $Z$ is obtained. I think most of the paper is fairly clearly written.

One thing that would help clarity is some more discussion on some of the mathematical aspects. For example the connections between isometry invariant kernels and densities is important to the paper (as we use this density to sample the eigenvalues for the eigenfunctions used), however, this discussion is terse.

in particular, there seems to be an assumption that every such kernel has a such a density associated with it. If this is known a reference would be appreciated.

**Quality**
---

I think the work is highly original and is of good quality.

**Reproducibility**
---

Again baring the issue of how the data is embedded I think the method is fairly reproducible.


**Questions**
---

1) The given pipeline can be used with *any data* and then *any network architecture* right? Is there a reason GNNs were chosen rather than DNNs?

2) I am curious about the estimation part of the paper and the number of eigenfunction used. Is there some sort of concentration happening here where as we increase the number of eigenfunction used the random variable $\langle \phi(x), \phi(y) \rangle$ concentrates to the mean? If this is the case, would the authors know how quickly it concentrates?

**Strength And Weaknesses:**

**Strengths**
---

1) The Random Features Model is very important model for Euclidean data and the extension to the Hyperbolic version seems fundamental and interesting.
2) The idea to then get Euclidean features that represent hyperbolic geometry (in some way) in this manner is very novel.
3) The theory results are compelling and general to show that this can be used in a wide variety of settings.

**Weaknesses**
---

1) Since the method for embedding the data into hyperbolic space is crucial to the method, the paper is missing a discussion on how to obtain this embedding. On page 6 it says ``A low dimensional hyperbolic embedding $z \in \mathcal{B}^{d_0}$ is initialized for each node to derive hyperbolic embeddings $Z \in \mathbb{R}^{n\times d_0}$ for all nodes in the graph''. However, this does not tell us how the embeddings was obtained. I think this is a crucial detail. Further, it is not clear, whether $Z$ is now a parameter is that learned during the neural network training. Based on the motivation of the paper to avoid hyperbolic gradient based learning, I do not think this is the case.

Hence I think some discussion of how the method learns the embeddings and a discussion (such as Nickel and Kiela NeurIPS 2017, Nickel and Kiela ICML 2018, Sala, De Sa, Gu and Re ICML 2018, Sonthalia and Gilbert NeurIPS 2020) about embedding data into hyperbolic space would help.

**Summary Of The Paper:**

This paper presents a new method for utilizing hyperbolic geometry that avoids many of the issues with current models.

My understanding of it is as follows. Suppose we have $x_1, \ldots, x_n$ embedded into some hyperbolic space and some kernel map $k(x_i,x_j)$ that **only** depends on the hyperbolic distance between the data, then we can define a random features type map (using the eigenfunctions of the Laplace Beltrami Operator on the manifold) to get features $\phi(x_i) \in \mathbb{R}^d$ such that $\mathbb{E}[\phi(x_i)^T \phi(x_j)] = k(x_i,x_j)$.

The idea then is to use this features with traditional neural networks (instead of hyperbolic neural networks). The paper proves that any such kernel can be estimated in an unbiased banner and implements one such kernel to show improvements in many cases over both purely hyperbolic and purely euclidean methods.

**Summary Of The Review:**

In summary I think the paper presents a fundamental extension of the random features model of hyperbolic space. It then presents a very interesting way in which this can be used.

I think the paper is novel, and interesting.

---

> ### Author Response · Authors · 2022-11-18
> **Response to Reviewer Ue38**
>
> We thank the reviewer for acknowledging our main contributions as novel and fundamental to the area with a nice summary! Please see below for responses to your questions and suggestions:
>
> Q1: how the hyperbolic embeddings $Z$ were obtained?
>
> We updated Sec. 5 of the paper to make it clear and included discussion on this in Appendix D. For the purpose of end-to-end learning, in our experiments, we jointly learn the embedding parameter $Z$ and weight $W$ in SGC during the training time. It's possible to adopt a two-step approach: first pretrain a hyperbolic embedding, then fix the embedding and train the graph learning model only. In the first step, optimization-based methods (Nickel and Kiela 2017, 2018) or combinatorial construction methods (Sala, De Sa, Gu and Re 2018, Sonthalia and Gilbert 2020) can be adopted by supervising the graph connectivity. However, these methods only utilize the graph structure information, but ignore the node feature information $X$, which leads to an obvious performance degradation. In comparison, as analyzed in Appendix C, our end-to-end learning of HyLa can be used to embed features and enables learning a complex kernel representation to avoid this shortcoming. Intuitively, the graph connectivity information can be too general for downstreaming tasks which rely more on semantic information. It's not clear to us how to encode the semantic information (node features) into embedding following e.g., Nickel and Kiela 2017, 2018. Please check Appendix D for more discussion.
>
> Q2: The given pipeline can be used with any data and then any network architecture right? Is there a reason GNNs were chosen rather than DNNs?
>
> The given pipeline can be used with any data and then any network architecture. However, note that if the input data is Euclidean (e.g., images), an appropriate mapping from Euclidean space to hyperbolic space is required (e.g., exponential map) before the random feature mapping can be used. In the graph setting, there’s no such need because the model forward starts from hyperbolic embeddings. Furthermore, hyperbolic space is shown to outperform Euclidean space in some graph applications, so we adopted GNNs rather than DNNs.
>
> Q3: Estimation part and concentration.
>
> For the estimation in Figure 3, we sampled $1,000$ different eigenfunctions (also noted in the caption and paper). There is concentration happening as we increase the number of eigenfunctions, we have updated the paper and included a discussion in Appendix B. Unfortunately, the eigenfunction itself is not a sub-Gaussian so as to derive the concentration bound in a straightforward manner, but we do provide a numerical experiment to measure the estimation behavior in Figure 6, which seems to be an exponentially decay. It's an interesting future work to investigate this estimation error.
>
> Question and suggestions in Clarity:
>
> Due to page limits, we can only add/modify part of discussions on the mathematical aspects. Our Theorem 4.2 holds supposing the existence of an associated density with the kernel, i.e., the existence of the spherical transform, which is included in the theorem and proof.
>
> Our code is provided in the supplementary material at submission time.

---

> > ### Comment · Reviewer_Ue38 · 2022-11-18
> > **Thanks for the clarifications**
> >
> > Thank you for the clarifications. I am still of the opinion that this is a novel and interesting paper that should be accepted.

---

### Official Review · Reviewer_iAHJ · 2022-10-25

**Confidence:** 5
**Clarity, Quality, Novelty And Reproducibility:** The paper is clear and easy to read.
**Correctness:** 3
**Technical Novelty And Significance:** 2
**Empirical Novelty And Significance:** 2
**Recommendation:** 5

**Strength And Weaknesses:**

The idea is quite interesting, but the experimental results are strange, especially for the baselines in table 1 which are much lower than their real performance.
For scalability, the author only experiments on some small datasets.

**Summary Of The Paper:**

In this paper, the authors suggest a simpler method: learn a hyperbolic embedding of the input, map it once to Euclidean space using a mapping that encodes geometric priors by respecting the isometries of hyperbolic space, and end with a standard Euclidean network.

**Summary Of The Review:**

In this paper, the authors propose a straightforward approach, learn a hyperbolic embedding of the input, then map it once to Euclidean space using a mapping that preserves the isometries of hyperbolic space while encoding geometric priors, and finally, finish with a standard Euclidean network. However, I am a little concerned about the experimental results.

---

> ### Author Response · Authors · 2022-11-18
> **Response to Reviewer iAHJ**
>
> Thanks for reviewing the paper, we are glad that you think our paper is “clear and easy to read”, please see below for answers to your questions and concerns:
>
> Q1: Experimental results are strange, especially for the baselines in table 1 which are much lower than their real performance.
>
> We don’t know which baseline experiment the reviewer is questioning about, but we provide a detailed breakdown below for all baseline experiments in Table 1,
>
> – For the HGCN [1] baseline, we took the reported results from their original paper.
>
> – For the GCN [2], HNN [3] baseline, we took the implementation from the HGCN paper and made multiple runs. The results are not only consistent with the reported results in the original paper, but also with later work replicating GCN, HNN as baselines, e.g., HGCN [1], GIL [4].
>
> – For SGC [5], GAT [6] on standard citation networks (cora, citeseer and pubmed), we took the reported results from their original paper. For SGC,GAT on new hyperbolic datasets (disease and airport proposed in HGCN), we took HGCN’s implementation [1] and made multiple runs. The results are consistent with the reported results of later work replicating them as baselines, e.g., HGCN [1], GIL [4]
>
> – For LGCN [7], the original implementation trained the model for thousand epochs and used early stopping with a patience of 100 epochs (noted in their original paper for the reported results). We mentioned in our baseline subsection that Euclidean baselines, e.g. GCN, SGC, GAT reported results of training the model for a maximum of 100 epochs, thus, for a fair comparison, we took the official code of LGCN, trained the model for 100 epochs and tuned hyperparameters in the same way as the original paper did. The results were averaged over multiple runs.
>
> [1] Hyperbolic graph convolutional neural networks, with implementation in https://github.com/HazyResearch/hgcn.
>
> [2] Semi-Supervised Classification with Graph Convolutional Networks
>
> [3] Hyperbolic neural networks
>
> [4] Graph Geometry Interaction Learning
>
> [5] Simplifying Graph Convolutional Networks
>
> [6] Graph Attention Networks
>
> [7] Lorentzian Graph Convolutional Networks, with implementation in https://github.com/ydzhang-stormstout/LGCN/tree/main/lgcn_torch
>
> Q2: For scalability, the author only experiments on some small datasets.
>
> For the choice of datasets, we followed previous work’s choices rather than new datasets that were not considered before so as to compare with them. Just for the reviewer’s reference, take the node classification task for example, we used Disease, Airport, Cora, Pubmed, Citeseer and Reddit datasets, which contains nearly all datasets adopted in previous work: GCN (Cora, Pubmed and Citeseer), SGC (Cora, Pubmed, Citeseer and Reddit), GAT (Cora, Pubmed, Citeseer and PPI, an inductive dataset contains multiple graphs), HGCN (Disease, Airport, Cora, Pubmed, PPI), LGCN (Disease, Cora, Pubmed, Citeseer, Amazon, USA), GIL (Disease, Airport, Cora, Pubmed, Citeseer). In particular, we included the inductive Reddit dataset which consists of a much larger graph, i.e., 233k nodes and 11.6M edges. For datasets on text classification, we followed previous work’s choice (e.g. SGC, textGCN [8]).
>
> [8] Graph Convolutional Networks for Text Classification

---

> > ### Comment · Reviewer_iAHJ · 2022-11-25
> > **The author's response is much appreciated.**
> >
> > The author's response is much appreciated. There are two concerns from me: experiments and motivation.
> >
> >
> > **1. Experiments: The comparisons are tricky--underfitting comparisons are fruitless.**
> >
> > Since the author's experimental settings are confusing and misleading, which is unfavorable to the development of the community, based on this, I do not recommend that the paper be accepted at the current time. It is advised that the author conduct a reasonable and mature comparison before considering the publication of the paper.
> >
> > For example, for the same dataset, Disease, Airport, PubMed, Citeseer, and Cora, the author's comparison is somewhat tricky and problematic. According to the author's description, the training epochs of the baseline is 100, while the HGCN is 5000 epochs with 100 patience. Also, according to Table 5, there are different training epochs for different datasets.
> >
> > There are several issues here:
> > - (1) There is a large discrepancy in baseline comparisons. Other baselines are trained with a max of 100 epochs while HGCN is 50000 epochs with 100 patience. What makes them different?
> > - (2) The training strategies of baselines are quite different from the method of the original paper.  For example, the basic settings of LGCN are similar to that of HGCN, but the author only trained a max of 100/(5000 epoch with 200 patience), and the results obtained are far lower than the results in LGCN.
> > - (3) The comparison in a state of far underfitting does not have much significance for the development of the community.
> > The main issue is that the baselines only trained with 100 epochs often do not fit the data well, and the comparisons of far underfitting results are meaningless. When the baselines are well-trained, they are much better than the author's study.
> >
> >
> > - (4) The author's reproduced results are much lower than their real performance. The followings are two examples:
> > - For the disease results from LGCN's performance, according to LGCN's official code, even with 100 training epochs (using the simple command: `python train.py --task nc --dataset disease_nc --model LGCN --lr 0.01 --dim 16  -- act relu --bias 1 --dropout 0 --weight-decay 0 --manifold Lorentzian --normalize-feats 1 --log-freq 5 --epochs 100`, where the number of layers is equal to the order in author’s code on the disease), similarly, we can obtain much better results with LGCN. Without parameter tuning, we can get impressive results with test_acc: 0.8702 and test_f1: 0.8291. If it is a fitting state with more training epochs, the author's method cannot be compared.
> >
> >
> > - For the airport results from LGCN's performance, similarly, according to LGCN's official code,  we can obtain much better results. The extremely poor and wrong results are harmful to the development of this community since the follower will not judge and claim that they follow this work.
> > Intuitively, there is not much difference between LGCN and HGCN, and LGCN generally obtains better results than HGCN since it constrains the embeddings always in hyperbolic geometry while HGCN cannot. However, in this work, the accuracy performance of LGCN is 57.6%, while the performance of HGCN is 90.6, which is hilarious and amazing. For the author's results, I guess the author utilizes the wrong datasets or data processing. Please check carefully that the dataset used is the airport, and that data processing is the same as HGCN.
> >
> > In addition, there are many mistakes in the author's understanding.
> >
> > - The standard GCN default training epochs is 200, and please refer to section 5.2 of https://arxiv.org/pdf/1609.02907.pdf
> > - The GAT default training epochs is 100000 with patience 100 (please refer to the following official code)
> > https://github.com/PetarV-/GAT/blob/master/execute_cora.py
> >
> > - The code of GIL is with detailed parameters, and the results of GIL are easily reproducible, https://github.com/CheriseZhu/GIL. If you have any questions, feel free to contact the author of the paper.
> >
> > Although the above training methods are slightly different, it is **undeniable** that they have all reached the state of fitting rather than underfitting.
> >
> > For time comparison, does the author consider **the calculation of eigenvalues** and the **high-order calculation** of the power of the adj matrix? How about the time on a large graph with billion-level nodes? The scalability of Hypla to large-scale datasets is not widely studied. There are still a number of problems with the scalability, such as the accessing adj matrix and aggregation problem in large-scale graph data. If SGC or the proposed method is scalable, then hyperbolic+SGC can also be scalable.
> >
> > **2.Motivation**
> >
> > - The author's proposal can be written as $softmax(A^k\tilde{X}\bar{X}W)$, since W is also learnable. It cannot come to a conclusion that the “improvement” is totally brought by $\bar{X}$.
> >
> > - How about we utilize the eigenfunctions of the Laplace-Beltrami operator for the trainable weights rather than the feature?

---

> > > ### Author Response · Authors · 2022-12-01
> > > **Author response part 1**
> > >
> > > Thanks for the response.
> > >
> > > – Experiment
> > >
> > > The reviewer kept criticizing our experiments, particularly baselines. However, from the author’s point of view, the reviewer is mostly questioning the $\textbf{LGCN}$ baseline, particularly that it is trained for 100 epochs rather than thousands epochs. As such, the reviewer believes that baselines are all underfitting. However, as mentioned in our last response, our baseline experiment results (apart from $\textbf{LGCN}$) are either taken from the original paper or consistent with later work replicating them as baselines. We address the reviewer’s questions in the following:
> > >
> > > (1) We pointed it clear in the paper that for baseline models, HGCN is the only exception trained for 5,000 epochs, because we studied on the same set of datasets as HGCN and we took their reported results, which is trained for 5,000 epochs with 100 patience in the original repo (https://github.com/HazyResearch/hgcn/blob/a526385744da25fc880f3da346e17d0fe33817f8/config.py#L10, and we don’t know how the 50,000 epoch number came from in the reviewer’s question?). For $\textbf{LGCN}$, experiments on some datasets were not conducted in the original paper, e.g., Airport, so we took their implementation and trained for 100 epochs as other baselines and our HyLa-SGC do. For the detailed hyper-parameters, we also note in the paper that please check more experimental details provided in Appendix E, e.g. Table 5, where you can observe a 200 epoch training for pubmed and 100 epochs for the rest datasets
> > >
> > > (2) The reviewer again questioned the training strategies of (all?) “baselines”, however, the only different one is LGCN. FYI, we will provide experiments with LGCN trained in the same way as HGCN in later responses.
> > >
> > > (3) As mentioned above and also in our last response, our baseline experiment results (apart from $\textbf{LGCN}$) are either taken from the original paper or consistent with later work replicating them as baselines. The reviewer made a statement that models trained for 100 epochs training do not fit the data well, does the reviewer have any evidence for this statement? Or is it only for the $\textbf{LGCN}$ baseline again? To the author’s knowledge, many Euclidean graph networks such as GCN, SGC can converge well in 100 or 200 epochs. If a proposed model cannot converge well within hundred epochs, this is already a disadvantage, while our HyLa-SGC can perform well within 100/200 epochs.
> > >
> > > (4) The reviewer seems to make a general criticism of our all baseline experiments, but only able to list $\textbf{LGCN}$ again as an example.
> > >
> > > —   For the disease dataset, it’s a small dataset and large variances can be observed. It’s possible to get a high performance with a single run. We made multiple runs of $\textbf{LGCN}$ trained with 100 epochs and reported a result 81.3±4.0, in LGCN original paper, when trained for thousand epochs with 100 patience, they reported a result 84.4±0.8. As you can see, both are lower than the reviewer’s 87.02 single run experiment, we suggest the reviewer make multiple runs with different seeds instead of deriving an immature conclusion easily.
> > >
> > > —  The Airport is again a small dataset and large variances can be observed with different seeds. We don’t know what numbers the reviewer is getting, but from our experience, with some particular random seeds, you can get 85+% test accuracy while with most seeds, a ~50% test accuracy is always derived. This dataset (proposed in HGCN) is not considered in the original $\textbf{LGCN}$ paper, where the other dataset (Disease) proposed in HGCN is studied. We suggest the reviewer make multiple runs with different seeds. At last, we took our datasets from HGCN repo and actually used the same set of loading utils as them. Rigorously speaking, the author do not agree with the reviewer that LGCN generally performs better than HGCN, particularly on a new dataset that is not considered in $\textbf{LGCN}$ original paper. The authors would appreciate it if the reviewer could provide evidence or proof for this statement?
> > >
> > > We thank the reviewer for pointing out our mistakes in some understandings, e.g., that GCN trained the model for 200 epochs (rather than thousand epochs with 100 patience :). For GIL baseline, we are not sure if the reviewer tried it or not, but it’s not easily reproducible as the reviewer imagined. Many environmental packages are out of date and not available, particularly the geoopt==0.0.1, torch_scatter == 1.3.0 and torch_geometric == 1.3.0. Even some functions in geoopt=0.0.1 are not available for call. We are contacting the authors to solve this, that’s why it’s not included as one baseline at submission time.

---

> > > ### Author Response · Authors · 2022-12-01
> > > **Author response part 2**
> > >
> > > The reviewer then arrives at a general conclusion that our experiments are underfitting, again our experimental results (except LGCN) are either taken from their original paper or consistent with later work replicating them as baselines. Overall, the reviewer posted questions nearly all related to the LGCN baseline, which is only because it is trained for 100 epochs rather than more. In order to solve this problem, we will conduct experiments of training LGCN for thousands of epochs with 100 epoch patience. The authors are currently attending a conference, please allow some time for the experiment result response to come, thank you.
> > >
> > > According to the reviewer’s rest questions, the authors find $\textbf{the reviewer do not fully understand our paper}$:
> > >
> > > – For the time comparison part, does the author consider the calculation of eigenvalues and the high-order calculation of the power of the adj matrix?
> > >
> > > HyLa adopts the same sampling technique as random Fourier features, as shown crystally clear in section 5, end of page 6, we sample the eigenvalues from a zero-mean normal distribution. Also mentioned in section 6 Efficiency part, we “taking into account the precomputation time of the models into training time”, the adj matrix precomputation in SGC and HyLa-SGC is also included. The adj order is either 2 or 5, rather than a high order number as shown in table 5.
> > >
> > > – How about the time on a large graph with billion-level nodes? The scalability of Hypla to large-scale datasets is not widely studied.
> > >
> > > In this work, we propose HyLa as an isometry-preserving feature mapping from hyperbolic space to Euclidean space. HyLa can be used with any Euclidean graph networks, based on which, we adopt HyLa with simple SGC to use hyperbolic space in a different way to previous work for graph learning tasks. We measured HyLa-SGC in both standard node classification and text classification datasets, including the very large scale reddit dataset with 233k nodes and 11.6M edges. We think it’s not an obligation and unreasonable for us to experiment on billion-level nodes graph datasets. Even so, as used in our experiments and discussed in first 2 paragraphs of page 7, HyLa is used at feature level, and its computational complexity is only related to the feature dimension and independent of the graph size, in which sense, HyLa works well theoretically for billion-level nodes graph.
> > >
> > > – Scalability: accessing adj matrix and aggregation problem in large-scale graph data. If SGC or the proposed method is scalable, then hyperbolic+SGC can also be scalable.
> > >
> > > The SGC is the most scalable graph network compared to GCN, HGCN etc.. Again we are not proposing a new graph network, we propose a new way to use hyperbolic space with simple SGC for graph learning, and HyLa-SGC scales well just as SGC does.
> > >
> > >
> > > – Motivation
> > > The author's proposal can be written as $softmax(A^kX\hat{X}W)$, since W is also learnable. It cannot come to a conclusion that the “improvement” is totally brought by $\hat{X}$.
> > >
> > > SGC can be written as $softmax(A^kXW)$, and our HyLa is a feature that is used together with some models, e.g. SGC, so HyLa-SGC is $softmax(A^kX\hat{X}W)$. We never claimed that the “improvement” is totally brought by $\hat{X}$, but we concluded that since HyLa-SGC (with both $\hat{X}, W$ learnt) outperforms SGC (with only $W$ learnt), then the $\hat{X}$ is useful and beneficial for the learning by a simple control of variable.
> > >
> > > – How about we utilize the eigenfunctions of the Laplace-Beltrami operator for the trainable weights rather than the feature?
> > >
> > > First of all, our proposed HyLa functions are eigenfunction of the Laplace-Beltrami operator. Secondly, the eigenfunctions of the Laplace-Beltrami operator is a mapping from hyperbolic space to Euclidean space, the trainable weights in our model, e.g., $W$ is an Euclidean weight, so you can not apply the eigenfunctions directly on it. In our method, we only used HyLa from the hyperbolic embeddings to Euclidean features as a new simple way to use hyperbolic space, different to previous methods with each layer weights in hyperbolic space.
> > >
> > > Again, we appreciate your time reviewing our paper, but clearly the reviewer doesn't fully understand our paper and criticize harshly on our LGCN baseline and blame blindly to all our experiments, simply because LGCN is not trained for thousand epochs. We will run experiments with LGCN trained for thousand epochs just to address the reviewer's question.

---

> > > ### Author Response · Authors · 2022-12-07
> > > **Author response part 3**
> > >
> > > Thanks for your patience, we made a thorough study of the LGCN baseline experiments. We folk LGCN/lgcn_torch GitHub repo, and train the LGCN model with the same config as HGCN (https://github.com/HazyResearch/hgcn/blob/master/config.py), specifically with a maximum of 5,000 epochs, minimum of 100 epochs and 100 epoch patience. We also observed an obvious difference when dropout is turned on (though HyLa-SGC does not use dropout), please see results below:
> > >
> > > When dropout=0.5 || dropout=0.0
> > >
> > > Disease: test acc 0.7835$\pm$0.0956, test F1 0.7461$\pm$0.0611   ||   test acc 0.7602$\pm$0.0960, test F1 0.7349$\pm$0.0467
> > >
> > > Airport: test acc 0.5874$\pm$0.0344, test F1 0.3814$\pm$0.0728   ||   test acc 0.8229$\pm$0.0184, test F1 0.7779$\pm$0.0291
> > >
> > > Pubmed: test acc 0.7712$\pm$0.0056   ||   test acc 0.7638$\pm$0.0081
> > >
> > > Citeseer: test acc 0.6352$\pm$0.0187   ||   test acc 0.6143$\pm$0.0195
> > >
> > > Cora: test acc 0.7814$\pm$0.0010   ||   test acc 0.7523$\pm$0.0127
> > >
> > > $\textbf{Under all cases, LGCN (trained with a maximum of 5000 epochs and 100 patience)}$
> > > $\textbf{underperforms our HyLa-SGC (trained for 100 epochs) results reported in the paper.}$
> > >
> > > More details:
> > >
> > > We tune the learning rate in the range 0.01, 0.008, 0.005, 0.001, use weight decay=0.0, dropout=0.0 or 0.5. All results are averaged over 10 runs. For example on Cora, we use the following command:
> > >
> > > python train.py --task nc --dataset cora --model LGCN --lr 0.008 --dim 16 --num-layers 2 --act relu --bias 1 --dropout 0.0 --weight-decay 0 --manifold Lorentzian --normalize-feats 1 --log-freq 5 --seed 43
> > >
> > > For the airport dataset, LGCN doesn't provide loading utils, so we took the dataset (https://github.com/HazyResearch/hgcn/tree/master/data/airport) and utils function (https://github.com/HazyResearch/hgcn/blob/a526385744da25fc880f3da346e17d0fe33817f8/utils/data_utils.py#L244) from HGCN repo.
> > > For early stopping, depending on the metric (test acc or f1), we use the rule m1["f1"] < m2["f1"] or m1["acc"] < m2["acc"] accordingly in (https://github.com/BUPT-GAMMA/lgcn_torch/blob/dd7560fb76b533d4241c99cd1e826cf90de10955/models/base_models.py#L90).
> > >
> > > For tuned hyperparameters:
> > >
> > > Disease: when dropout=0.0, lr 0.005; when dropout=0.5, lr 0.005
> > >
> > > Airport: when dropout=0.0, lr 0.01; when dropout=0.5, lr 0.008
> > >
> > > Pubmed: when dropout=0.0, lr 0.008; when dropout=0.5, lr 0.008
> > >
> > > Citeseer: when dropout=0.0, lr 0.005; when dropout=0.5, lr 0.001
> > >
> > > Cora: when dropout=0.0, lr 0.008; when dropout=0.5, lr 0.005
> > >
> > > We can also provide detailed performance statistics for each run and each dataset if needed. Overall, our HyLa-SGC outperforms LGCN easily in all cases. Many claims and criticisms of the reviewer are absurd and lack of supporting evidence. We hope the meta reviewer can consider and judge this review seriously and reasonably. We are welcome to any discussion.

---

> ### Comment · Reviewer_iAHJ · 2022-12-09
> **Thanks for the author's response.**
>
> The author's method is interesting, but the experimental part still needs to be improved and updated. However, at this moment, it cannot be modified any further.
>
> Much appreciated the author's response. I am raising my score to 5.
>
> It can fully be accepted on my side if the author's final version of the experiment is reasonable and fair. @Area Chair Wk4e

---

### Author Response · Authors · 2022-11-18
**Response to All Reviewers**

We thank all the reviewers for their valuable comments and suggestions on our work. We are glad that reviewers find our paper clear and easy to follow (R1 [iAHJ], R3 [BVuW]). We are pleased that R2 [Ue38], R3 [BVuW] acknowledge our work as novel and fundamental contributions to the area.

We provided detailed responses to each review and also revised our manuscript following the suggestions or in correspondence to some questions. Our code is provided in the supplementary material at the first time of submission. We summarize the major changes in the paper below:

1. We updated Sec. 5 to make it clear on how we derive the hyperbolic embedding and also include more discussions on this in both Sec. 5 and Appendix D.

2. We included more discussions and experiments on the estimation error/concentration behavior in Appendix B.

3. We further illustrate the usage of HyLa at node level and feature level in Appendix C.

Minor change: we modified Theorem 4.2 and its proof: 1) removed the $1/2$ factor to make the formula more clear and consistent; 2) included the existence condition of the associated density.

---

### Decision · Program_Chairs · 2023-01-20

**Decision:**

Accept: poster

**Justification For Why Not Higher Score:**

insufficient contribution

**Justification For Why Not Lower Score:**

cool idea

**Metareview: Summary, Strengths And Weaknesses:**

The paper proposes Hyperbolic Laplacian Features, a hyperbolic analogy of the classical Random Fourier Features of Rahimi (2007), and shows how they can be used to enhance learning on graphs (as a simple alternative to full hyperbolic GNNs). The method is interesting and efficient, and was appreciated by the reviewers. We recommend acceptance.



**Note From Pc:**

if the above contains the word "oral" or "spotlight" please see: "oral" presentation means -> notable-top-5% and "spotlight" means -> notable-top-25%. As stated in our emails, we are disassociating presentation type from AC recommendations